# On the Information Theoretic Limits of Learning Ising Models

**Karthikeyan Shanmugam**[1*]**, Rashish Tandon**[2†]**, Alexandros G. Dimakis**[1‡]**, Pradeep Ravikumar**[2*]
[1]Department of Electrical and Computer Engineering, [2]Department of Computer Science
The University of Texas at Austin, USA
[*]karthiksh@utexas.edu, [†]rashish@cs.utexas.edu
[‡]dimakis@austin.utexas.edu, [*]pradeepr@cs.utexas.edu

## Abstract

We provide a general framework for computing lower-bounds on the sample complexity of recovering the underlying graphs of Ising models, given $i.i.d.$ samples. While there have been recent results for specific graph classes, these involve fairly extensive technical arguments that are specialized to each specific graph class. In contrast, we isolate two key graph-structural ingredients that can then be used to specify sample complexity lower-bounds. Presence of these structural properties makes the graph class hard to learn. We derive corollaries of our main result that not only recover existing recent results, but also provide lower bounds for novel graph classes not considered previously. We also extend our framework to the random graph setting and derive corollaries for Erdős-Rényi graphs in a certain dense setting.

## 1  Introduction

Graphical models provide compact representations of multivariate distributions using graphs that represent Markov conditional independencies in the distribution. They are thus widely used in a number of machine learning domains where there are a large number of random variables, including natural language processing [13], image processing [6, 10, 19], statistical physics [11], and spatial statistics [15]. In many of these domains, a key problem of interest is to recover the underlying dependencies, represented by the graph, given samples *i.e.* to estimate the graph of dependencies given instances drawn from the distribution. A common regime where this graph selection problem is of interest is the *high-dimensional* setting, where the number of samples $n$ is potentially smaller than the number of variables $p$. Given the importance of this problem, it is instructive to have *lower bounds* on the sample complexity of any estimator: it clarifies the statistical difficulty of the underlying problem, and moreover it could serve as a certificate of optimality in terms of sample complexity for any estimator that actually achieves this lower bound. We are particularly interested in such lower bounds under the structural constraint that the graph lies within a given class of graphs (such as degree-bounded graphs, bounded-girth graphs, and so on).

The simplest approach to obtaining such bounds involves graph counting arguments, and an application of Fano's lemma. [2, 17] for instance derive such bounds for the case of degree-bounded and power-law graph classes respectively. This approach however is purely graph-theoretic, and thus fails to capture the interaction of the graphical model parameters with the graph structural constraints, and thus typically provides suboptimal lower bounds (as also observed in [16]). The other standard approach requires a more complicated argument through Fano's lemma that requires finding a subset of graphs such that (a) the subset is large enough in number, and (b) the graphs in the subset are close enough in a suitable metric, typically the KL-divergence of the corresponding distributions. This approach is however much more technically intensive, and even for the simple

classes of bounded degree and bounded edge graphs for Ising models, [16] required fairly extensive arguments in using the above approach to provide lower bounds.

In modern high-dimensional settings, it is becoming increasingly important to incorporate structural constraints in statistical estimation, and graph classes are a key interpretable structural constraint. But a new graph class would entail an entirely new (and technically intensive) derivation of the corresponding sample complexity lower bounds. In this paper, we are thus interested in isolating the key ingredients required in computing such lower bounds. This key ingredient involves one the following structural characterizations: (1) connectivity by short paths between pairs of nodes, or (2) existence of many graphs that only differ by an edge. As corollaries of this framework, we not only recover the results in [16] for the simple cases of degree and edge bounded graphs, but to several more classes of graphs, for which achievability results have already been proposed[1]. Moreover, using structural arguments allows us to bring out the dependence of the edge-weights, $\lambda$, on the sample complexity. We are able to show same sample complexity requirements for $d$-regular graphs, as is for degree $d$-bounded graphs, whilst the former class is much smaller. We also extend our framework to the random graph setting, and as a corollary, establish lower bound requirements for the class of Erdős-Rényi graphs in a dense setting. Here, we show that under a certain scaling of the edge-weights $\lambda$, $\mathcal{G}_{p,c/p}$ requires exponentially many samples, as opposed to a polynomial requirement suggested from earlier bounds[1].

## 2    Preliminaries and Definitions

**Notation:** $\mathbb{R}$ represents the real line. $[p]$ denotes the set of integers from 1 to $p$. Let $\mathbf{1}_S$ denote the vector of ones and zeros where $S$ is the set of coordinates containing 1. Let $A - B$ denote $A \bigcap B^c$ and $A\Delta B$ denote the symmetric difference for two sets $A$ and $B$.

In this work, we consider the problem of learning the graph structure of an Ising model. Ising models are a class of graphical model distributions over binary vectors, characterized by the pair $(G(V,E), \bar{\theta})$, where $G(V,E)$ is an undirected graph on $p$ vertices and $\bar{\theta} \in \mathbb{R}^{\binom{p}{2}} : \bar{\theta}_{i,j} = 0 \ \forall (i,j) \notin E$, $\bar{\theta}_{i,j} \neq 0 \ \forall \ (i,j) \in E$. Let $\mathcal{X} = \{+1, -1\}$. Then, for the pair $(G, \bar{\theta})$, the distribution on $\mathcal{X}^p$ is given as: $f_{G,\bar{\theta}}(\mathbf{x}) = \frac{1}{Z} \exp \left( \sum_{i,j} \bar{\theta}_{i,j} \, x_i x_j \right)$ where $\mathbf{x} \in \mathcal{X}^p$ and $Z$ is the normalization factor, also known as the *partition function*.

Thus, we obtain a family of distributions by considering a set of edge-weighted graphs $\mathcal{G}_\theta$, where each element of $\mathcal{G}_\theta$ is a pair $(G, \bar{\theta})$. In other words, every member of the class $\mathcal{G}_\theta$ is a weighted undirected graph. Let $\mathcal{G}$ denote the set of distinct unweighted graphs in the class $\mathcal{G}_\theta$.

A learning algorithm that learns the graph $G$ (and not the weights $\bar{\theta}$) from $n$ independent samples (each sample is a $p$-dimensional binary vector) drawn from the distribution $f_{G,\bar{\theta}}(\cdot)$, is an efficiently computable map $\phi : \chi^{np} \to \mathcal{G}$ which maps the input samples $\{\mathbf{x}_1, \ldots \mathbf{x}_n\}$ to an undirected graph $\hat{G} \in \mathcal{G}$ i.e. $\hat{G} = \phi(\mathbf{x}_1, \ldots, \mathbf{x}_n)$.

We now discuss two metrics of reliability for such an estimator $\phi$. For a given $(G, \bar{\theta})$, the probability of error (over the samples drawn) is given by $p(G, \bar{\theta}) = \Pr \left( \hat{G} \neq G \right)$. Given a graph class $\mathcal{G}_\theta$, one may consider the maximum probability of error for the map $\phi$, given as:

$$p_{\max} = \max_{(G,\theta) \in \mathcal{G}_\theta} \Pr \left( \hat{G} \neq G \right). \tag{1}$$

The goal of any estimator $\phi$ would be to achieve as low a $p_{\max}$ as possible. Alternatively, there are random graph classes that come naturally endowed with a probability measure $\mu(G, \theta)$ of choosing the graphical model. In this case, the quantity we would want to minimize would be the average probability of error of the map $\phi$, given as:

$$p_{\mathrm{avg}} = \mathbb{E}_\mu \left[ \Pr \left( \hat{G} \neq G \right) \right] \tag{2}$$

In this work, we are interested in answering the following question: For any estimator $\phi$, what is the minimum number of samples $n$, needed to guarantee an asymptotically small $p_{max}$ or $p_{avg}$ ? The answer depends on $\mathcal{G}_\theta$ and $\mu$(when applicable).

For the sake of simplicity, we impose the following restrictions[1]: We restrict to the set of *zero-field ferromagnetic* Ising models, where *zero-field* refers to a lack of node weights, and *ferromagnetic* refers to all positive edge weights. Further, we will restrict all the non-zero edge weights ($\bar{\theta}_{i,j}$) in the graph classes to be the same, set equal to $\lambda > 0$. Therefore, for a given $G(V, E)$, we have $\bar{\theta} = \lambda \mathbf{1}_E$ for some $\lambda > 0$. A deterministic graph class is described by a scalar $\lambda > 0$ and the family of graphs $\mathcal{G}$. In the case of a random graph class, we describe it by a scalar $\lambda > 0$ and a probability measure $\mu$, the measure being solely on the structure of the graph $G$ (on $\mathcal{G}$).

Since we have the same weight $\lambda (> 0)$ on all edges, henceforth we will skip the reference to it, i.e. the graph class will simply be denoted $\mathcal{G}$ and for a given $G \in \mathcal{G}$, the distribution will be denoted by $f_G(\cdot)$, with the dependence on $\lambda$ being implicit. Before proceeding further, we summarize the following additional notation. For any two distributions $f_G$ and $f_{G'}$, corresponding to the graphs $G$ and $G'$ respectively, we denote the Kullback-Liebler divergence (KL-divergence) between them as $D\left(f_G \| f_{G'}\right) = \sum_{x \in \mathcal{X}^p} f_G(x) \log\left(\frac{f_G(x)}{f_{G'}(x)}\right)$. For any subset $\mathcal{T} \subseteq \mathcal{G}$, we let $C_{\mathcal{T}}(\epsilon)$ denote an $\epsilon$-covering w.r.t. the KL-divergence (of the corresponding distributions) *i.e.* $C_{\mathcal{T}}(\epsilon)(\subseteq \mathcal{G})$ is a set of graphs such that for any $G \in \mathcal{T}$, there exists a $G' \in C_{\mathcal{T}}(\epsilon)$ satisfying $D\left(f_G \| f_{G'}\right) \leq \epsilon$. We denote the entropy of any r.v. $X$ by $H(X)$, and the mutual information between any two r.v.s $X$ and $Y$, by $I(X; Y)$. The rest of the paper is organized as follows. Section 3 describes Fano's lemma, a basic tool employed in computing information-theoretic lower bounds. Section 4 identifies key structural properties that lead to large sample requirements. Section 5 applies the results of Sections 3 and 4 on a number of different deterministic graph classes to obtain lower bound estimates. Section 6 obtains lower bound estimates for Erdős-Rényi random graphs in a *dense* regime. All proofs can be found in the Appendix (see supplementary material).

## 3 Fano's Lemma and Variants

Fano's lemma [5] is a primary tool for obtaining bounds on the average probability of error, $p_{avg}$. It provides a lower bound on the probability of error of any estimator $\phi$ in terms of the entropy $H(\cdot)$ of the output space, the cardinality of the output space, and the mutual information $I(\cdot, \cdot)$ between the input and the output. The case of $p_{max}$ is interesting only when we have a deterministic graph class $\mathcal{G}$, and can be handled through Fano's lemma again by considering a uniform distribution on the graph class.

**Lemma 1** (Fano's Lemma). *Consider a graph class $\mathcal{G}$ with measure $\mu$. Let, $G \sim \mu$, and let $X^n = \{\mathbf{x}_1, \ldots, \mathbf{x}_n\}$ be $n$ independent samples such that $\mathbf{x}_i \sim f_G$, $i \in [n]$. Then, for $p_{max}$ and $p_{avg}$ as defined in* (1) *and* (2) *respectively,*

$$p_{max} \geq p_{avg} \geq \frac{H(G) - I(G; X^n) - \log 2}{\log|\mathcal{G}|} \tag{3}$$

Thus in order to use this Lemma, we need to bound two quantities: the entropy $H(G)$, and the mutual information $I(G; X^n)$. The entropy can typically be obtained or bounded very simply; for instance, with a uniform distribution over the set of graphs $\mathcal{G}$, $H(G) = \log|\mathcal{G}|$. The mutual information is a much trickier object to bound however, and is where the technical complexity largely arises. We can however simply obtain the following loose bound: $I(G; X^n) \leq H(X^n) \leq np$. We thus arrive at the following corollary:

**Corollary 1.** *Consider a graph class $\mathcal{G}$. Then, $p_{max} \geq 1 - \frac{np + \log 2}{\log|\mathcal{G}|}$.*

**Remark 1.** *From Corollary 1, we get: If $n \leq \frac{\log|\mathcal{G}|}{p}\left((1 - \delta) - \frac{\log 2}{\log|\mathcal{G}|}\right)$, then $p_{max} \geq \delta$. Note that this bound on $n$ is only in terms of the cardinality of the graph class $\mathcal{G}$, and therefore, would not involve any dependence on $\lambda$ (and consequently, be very loose).*

To obtain sharper lower bound guarantees that depends on graphical model parameters, it is useful to consider instead a conditional form of Fano's lemma[1, Lemma 9], which allows us to obtain lower bounds on $p_{avg}$ in terms conditional analogs of the quantities in Lemma 1. For the case of $p_{max}$, these conditional analogs correspond to uniform measures on subsets of the original class $\mathcal{G}$.

The conditional version allows us to focus on potentially harder to learn subsets of the graph class, leading to sharper lower bound guarantees. Also, for a random graph class, the entropy $H(G)$ may be asymptotically much smaller than the log cardinality of the graph class, $\log|\mathcal{G}|$ (e.g. Erdős-Rényi random graphs; see Section 6), rendering the bound in Lemma 1 useless. The conditional version allows us to circumvent this issue by focusing on a *high-probability* subset of the graph class.

**Lemma 2** (Conditional Fano's Lemma). *Consider a graph class $\mathcal{G}$ with measure $\mu$. Let, $G \sim \mu$, and let $X^n = \{\mathbf{x}_1, \ldots, \mathbf{x}_n\}$ be $n$ independent samples such that $\mathbf{x}_i \sim f_G$, $i \in [n]$. Consider any $\mathcal{T} \subseteq \mathcal{G}$ and let $\mu(\mathcal{T})$ be the measure of this subset i.e. $\mu(\mathcal{T}) = \Pr_\mu(G \in \mathcal{T})$. Then, we have*

$$p_{avg} \geq \mu(\mathcal{T}) \, \frac{H(G|G \in \mathcal{T}) - I(G; X^n|G \in \mathcal{T}) - \log 2}{\log|\mathcal{T}|} \quad and,$$

$$p_{max} \geq \frac{H(G|G \in \mathcal{T}) - I(G; X^n|G \in \mathcal{T}) - \log 2}{\log|\mathcal{T}|}$$

Given Lemma 2, or even Lemma 1, it is the sharpness of an upper bound on the mutual information that governs the sharpness of lower bounds on the probability of error (and effectively, the number of samples $n$). In contrast to the trivial upper bound used in the corollary above, we next use a tighter bound from [20], which relates the mutual information to coverings in terms of the KL-divergence, applied to Lemma 2. Note that, as stated earlier, we simply impose a uniform distribution on $\mathcal{G}$ when dealing with $p_{max}$. Analogous bounds can be obtained for $p_{avg}$.

**Corollary 2.** *Consider a graph class $\mathcal{G}$, and any $\mathcal{T} \subseteq \mathcal{G}$. Recall the definition of $C_\mathcal{T}(\epsilon)$ from Section 2. For any $\epsilon > 0$, we have $p_{max} \geq \left(1 - \frac{\log|C_\mathcal{T}(\epsilon)| + n\epsilon + \log 2}{\log|\mathcal{T}|}\right)$.*

**Remark 2.** *From Corollary 2, we get: If $n \leq \frac{\log|\mathcal{T}|}{\epsilon} \left((1 - \delta) - \frac{\log 2}{\log|\mathcal{T}|} - \frac{\log|C_\mathcal{T}(\epsilon)|}{\log|\mathcal{T}|}\right)$, then $p_{max} \geq \delta$. $\epsilon$ is an upper bound on the radius of the KL-balls in the covering, and usually varies with $\lambda$.*

But this corollary cannot be immediately used given a graph class: it requires us to specify a subset $\mathcal{T}$ of the overall graph class, the term $\epsilon$, and the $KL$-covering $C_\mathcal{T}(\epsilon)$.

We can simplify the bound above by setting $\epsilon$ to be the radius of a single KL-ball w.r.t. some center, covering the whole set $\mathcal{T}$. Suppose this radius is $\rho$, then the size of the covering set is just 1. In this case, from Remark 2, we get: If $n \leq \frac{\log|\mathcal{T}|}{\rho} \left((1 - \delta) - \frac{\log 2}{\log|\mathcal{T}|}\right)$, then $p_{max} \geq \delta$. Thus, our goal in the sequel would be to provide a general mechanism to derive such a subset $\mathcal{T}$: that is large in number and yet has small diameter with respect to KL-divergence.

We note that Fano's lemma and variants described in this section are standard, and have been applied to a number of problems in statistical estimation [1, 14, 16, 20, 21].

## 4 Structural conditions governing Correlation

As discussed in the previous section, we want to find subsets $\mathcal{T}$ that are large in size, and yet have a small KL-diameter. In this section, we summarize certain structural properties that result in small KL-diameter. Thereafter, finding a large set $\mathcal{T}$ would amount to finding a large number of graphs in the graph class $\mathcal{G}$ that satisfy these structural properties.

As a first step, we need to get a sense of when two graphs would have corresponding distributions with a small KL-divergence. To do so, we need a general upper bound on the KL-divergence between the corresponding distributions. A simple strategy is to simply bound it by its symmetric divergence[16]. In this case, a little calculation shows :

$$D(f_G\|f_{G'}) \leq D(f_G\|f_{G'}) + D(f_{G'}\|f_G)$$
$$= \sum_{(s,t) \in E \setminus E'} \lambda \left(\mathbb{E}_G[x_s x_t] - \mathbb{E}_{G'}[x_s x_t]\right) + \sum_{(s,t) \in E' \setminus E} \lambda \left(\mathbb{E}_{G'}[x_s x_t] - \mathbb{E}_G[x_s x_t]\right)$$
$$(4)$$

where $E$ and $E'$ are the edges in the graphs $G$ and $G'$ respectively, and $\mathbb{E}_G[\cdot]$ denotes the expectation under $f_G$. Also note that the correlation between $x_s$ and $x_t$, $\mathbb{E}_G[x_s x_t] = 2P_G(x_s x_t = +1) - 1$.

From Eq. (4), we observe that the only pairs, $(s,t)$, contributing to the KL-divergence are the ones that lie in the symmetric difference, $E \Delta E'$. If the number of such pairs is small, and the difference of correlations in $G$ and $G'$ (*i.e.* $\mathbb{E}_G[x_s x_t] - \mathbb{E}_{G'}[x_s x_t]$) for such pairs is small, then the KL-divergence would be small.

To summarize the setting so far, to obtain a tight lower bound on sample complexity for a class of graphs, we need to find a subset of graphs $\mathcal{T}$ with small KL diameter. The key to this is to identify when KL divergence between (distributions corresponding to) two graphs would be small. And the key to this in turn is to identify when there would be only a small difference in the *correlations* between a pair of variables across the two graphs $G$ and $G'$. In the subsequent subsections, we provide two simple and general structural characterizations that achieve such a small difference of correlations across $G$ and $G'$.

### 4.1 Structural Characterization with Large Correlation

One scenario when there might be a small difference in correlations is when one of the correlations is very large, specifically arbitrarily close to 1, say $\mathbb{E}_{G'}[x_s x_t] \geq 1 - \epsilon$, for some $\epsilon > 0$. Then, $\mathbb{E}_G[x_s x_t] - \mathbb{E}_{G'}[x_s x_t] \leq \epsilon$, since $\mathbb{E}_G[x_s x_t] \leq 1$. Indeed, when $s, t$ are part of a clique[16], this is achieved since the large number of connections between them force a higher probability of agreement *i.e.* $P_G(x_s x_t = +1)$ is large.

In this work we provide a more general characterization of when this might happen by relying on the following key lemma that connects the presence of "many" node disjoint "short" paths between a pair of nodes in the graph to high correlation between them. We define the property formally below.

**Definition 1.** *Two nodes $a$ and $b$ in an undirected graph $G$ are said to be $(\ell, d)$ connected if they have $d$ node disjoint paths of length at most $\ell$.*

**Lemma 3.** *Consider a graph $G$ and a scalar $\lambda > 0$. Consider the distribution $f_G(\mathbf{x})$ induced by the graph. If a pair of nodes $a$ and $b$ are $(\ell, d)$ connected, then* $\mathbb{E}_G[x_a x_b] \geq 1 - \frac{2}{1 + \frac{(1+(\tanh(\lambda))^\ell)^d}{(1-(\tanh(\lambda))^\ell)^d}}$.

From the above lemma, we can observe that as $\ell$ gets smaller and $d$ gets larger, $\mathbb{E}_G[x_a x_b]$ approaches its maximum value of 1. As an example, in a $k$-clique, any two vertices, $s$ and $t$, are $(2, k-1)$ connected. In this case, the bound from Lemma 3 gives us: $\mathbb{E}_G[x_a x_b] \geq 1 - \frac{2}{1+(\cosh \lambda)^{k-1}}$. Of course, a clique enjoys a lot more connectivity (*i.e.* also $(3, \frac{k-1}{2})$ connected etc., albeit with node overlaps) which allows for a stronger bound of $\sim 1 - \frac{\lambda k e^\lambda}{e^{\lambda k}}$ (see [16])[2]

Now, as discussed earlier, a high correlation between a pair of nodes contributes a small term to the KL-divergence. This is stated in the following corollary.

**Corollary 3.** *Consider two graphs $G(V, E)$ and $G'(V, E')$ and scalar weight $\lambda > 0$ such that $E - E'$ and $E' - E$ only contain pairs of nodes that are $(\ell, d)$ connected in graphs $G'$ and $G$ respectively, then the KL-divergence between $f_G$ and $f_{G'}$,* $D(f_G \| f_{G'}) \leq \frac{2\lambda |E \Delta E'|}{1 + \frac{(1+(\tanh(\lambda))^\ell)^d}{(1-(\tanh(\lambda))^\ell)^d}}$.

### 4.2 Structural Characterization with Low Correlation

Another scenario where there might be a small difference in correlations between an edge pair across two graphs is when the graphs themselves are close in Hamming distance *i.e.* they differ by only a few edges. This is formalized below for the situation when they differ by only one edge.

**Definition 2** (Hamming Distance). *Consider two graphs $G(V, E)$ and $G'(V, E')$. The hamming distance between the graphs, denoted by $\mathcal{H}(G, G')$, is the number of edges where the two graphs differ* i.e.

$$\mathcal{H}(G, G') = |\{(s,t) \,|\, (s,t) \in E \Delta E'\}| \tag{5}$$

**Lemma 4.** *Consider two graphs $G(V, E)$ and $G'(V, E')$ such that $\mathcal{H}(G, G') = 1$, and $(a, b) \in E$ is the single edge in $E \Delta E'$. Then, $\mathbb{E}_{f_G}[x_a x_b] - \mathbb{E}_{f_{G'}}[x_a x_b] \leq \tanh(\lambda)$. Also, the KL-divergence between the distributions, $D(f_G \| f_G') \leq \lambda \tanh(\lambda)$.*

The above bound is useful in low $\lambda$ settings. In this regime $\lambda \tanh \lambda$ roughly behaves as $\lambda^2$. So, a smaller $\lambda$ would correspond to a smaller KL-divergence.

### 4.3 Influence of Structure on Sample Complexity

Now, we provide some high-level intuition behind why the structural characterizations above would be useful for lower bounds that go beyond the technical reasons underlying Fano's Lemma that we have specified so far. Let us assume that $\lambda > 0$ is a positive real constant. In a graph even when the edge $(s, t)$ is removed, $(s, t)$ being $(\ell, d)$ connected ensures that the correlation between $s$ and $t$ is still very high (exponentially close to 1). Therefore, resolving the question of the presence/absence of the edge $(s, t)$ would be difficult – requiring lots of samples. This is analogous in principle to the argument in [16] used for establishing hardness of learning of a set of graphs each of which is obtained by removing a *single* edge from a clique, still ensuring many short paths between any two vertices. Similarly, if the graphs, $G$ and $G'$, are close in Hamming distance, then their corresponding distributions, $f_G$ and $f_{G'}$, also tend to be similar. Again, it becomes difficult to tease apart which distribution the samples observed may have originated from.

## 5 Application to Deterministic Graph Classes

In this section, we provide lower bound estimates for a number of deterministic graph families. This is done by explicitly finding a subset $\mathcal{T}$ of the graph class $\mathcal{G}$, based on the structural properties of the previous section. See the supplementary material for details of these constructions. A common underlying theme to all is the following: We try to find a graph in $\mathcal{G}$ containing *many* edge pairs $(u, v)$ such that their end vertices, $u$ and $v$, have *many* paths between them (possibly, node disjoint). Once we have such a graph, we construct a subset $\mathcal{T}$ by removing one of the edges for these *well-connected* edge pairs. This ensures that the new graphs differ from the original in only the *well-connected* pairs. Alternatively, by removing any edge (and not just *well-connected* pairs) we can get another *larger* family $\mathcal{T}$ which is 1-hamming away from the original graph.

### 5.1 Path Restricted Graphs

Let $\mathcal{G}_{p,\eta}$ be the class of all graphs on $p$ vertices with have at most $\eta$ paths ($\eta = o(p)$) between any two vertices. We have the following theorem :

**Theorem 1.** *For the class* $\mathcal{G}_{p,\eta}$, *if* $n \leq (1 - \delta) \max \left\{ \frac{\log(p/2)}{\lambda \tanh \lambda}, \frac{1 + \cosh(2\lambda)^{\eta-1}}{2\lambda} \log \left( \frac{p}{2(\eta+1)} \right) \right\}$, *then* $p_{max} \geq \delta$.

To understand the scaling, it is useful to think of $\cosh(2\lambda)$ to be roughly exponential in $\lambda^2$ *i.e.* $\cosh(2\lambda) \sim e^{\Theta(\lambda^2)}$[3]. In this case, from the second term, we need $n \sim \Omega \left( \frac{e^{\lambda^2 \eta}}{\lambda} \log \left( \frac{p}{\eta} \right) \right)$ samples. If $\eta$ is scaling with $p$, this can be prohibitively large (exponential in $\lambda^2 \eta$). Thus, to have low sample complexity, we must enforce $\lambda = O(1/\sqrt{\eta})$. In this case, the first term gives $n = \Omega(\eta \log p)$, since $\lambda \tanh(\lambda) \sim \lambda^2$, for small $\lambda$.

We may also consider a generalization of $\mathcal{G}_{p,\eta}$. Let $\mathcal{G}_{p,\eta,\gamma}$ be the set of all graphs on $p$ vertices such that there are at most $\eta$ paths of length at most $\gamma$ between any two nodes (with $\eta + \gamma = o(p)$). Note that there may be more paths of length $> \gamma$.

**Theorem 2.** *Consider the graph class* $\mathcal{G}_{p,\eta,\gamma}$. *For any* $\nu \in (0, 1)$, *let* $t_\nu = \frac{p^{1-\nu} - (\eta+1)}{\gamma}$. *If* $n \leq$

$(1 - \delta) \max \left\{ \frac{\log(p/2)}{\lambda \tanh \lambda}, \frac{1 + \left[ \cosh(2\lambda)^{\eta-1} \left( \frac{1 + \tanh(\lambda)^{\gamma+1}}{1 - \tanh(\lambda)^{\gamma+1}} \right)^{t_\nu} \right]}{2\lambda} \nu \log(p) \right\}$, *then* $p_{max} \geq \delta$.

The parameter $\nu \in (0, 1)$ in the bound above may be adjusted based on the scaling of $\eta$ and $\gamma$. Also, an approximate way to think of the scaling of $\left( \frac{1 + \tanh(\lambda)^{\gamma+1}}{1 - \tanh(\lambda)^{\gamma+1}} \right)$ is $\sim e^{\lambda^{\gamma+1}}$. As an example, for constant $\eta$ and $\gamma$, we may choose $v = \frac{1}{2}$. In this case, for some constant $c$, our bound imposes $n \sim \Omega \left( \frac{\log p}{\lambda \tanh \lambda}, \frac{e^{c\lambda^{\gamma+1} \sqrt{p}}}{\lambda} \log p \right)$. Now, same as earlier, to have low sample complexity, we must

have $\lambda = O(1/p^{1/2(\gamma+1)})$, in which case, we get a $n \sim \Omega(p^{1/(\gamma+1)} \log p)$ sample requirement from the first term.

We note that the family $\mathcal{G}_{p,\eta,\gamma}$ is also studied in [1], and for which, an algorithm is proposed. Under certain assumptions in [1], and the restrictions: $\eta = O(1)$, and $\gamma$ is *large enough*, the algorithm in [1] requires $\frac{\log p}{\lambda^2}$ samples, which is matched by the first term in our lower bound. Therefore, the algorithm in [1] is optimal, for the setting considered.

## 5.2 Girth Bounded Graphs

The girth of a graph is defined as the length of its shortest cycle. Let $\mathcal{G}_{p,g,d}$ be the set of all graphs with girth atleast $g$, and maximum degree $d$. Note that as girth increases the learning problem becomes easier, with the extreme case of $g = \infty$ (*i.e.* trees) being solved by the well known Chow-Liu algorithm[3] in $O(\log p)$ samples. We have the following theorem:

**Theorem 3.** *Consider the graph class* $\mathcal{G}_{p,g,d}$. *For any* $\nu \in (0,1)$, *let* $d_\nu = \min\left(d, \frac{p^{1-\nu}}{g}\right)$. *If*

$$n \leq (1-\delta) \max \left\{ \frac{\log(p/2)}{\lambda \tanh \lambda}, \frac{1 + \left(\frac{1+\tanh(\lambda)^{g-1}}{1-\tanh(\lambda)^{g-1}}\right)^{d_\nu}}{2\lambda} \nu \log(p) \right\}, \text{ then } p_{max} \geq \delta.$$

## 5.3 Approximate $d$-Regular Graphs

Let $\mathcal{G}_{p,d}^{\text{approx}}$ be the set of all graphs whose vertices have degree $d$ or degree $d-1$. Note that this class is subset of the class of graphs with degree at most $d$. We have:

**Theorem 4.** *Consider the class* $\mathcal{G}_{p,d}^{\text{approx}}$. *If* $n \leq (1-\delta) \max \left\{ \frac{\log\left(\frac{pd}{4}\right)}{\lambda \tanh \lambda}, \frac{e^{\lambda d}}{2\lambda d e^\lambda} \left(\frac{pd}{4}\right) \right\}$ *then* $p_{max} \geq \delta$.

Note that the second term in the bound above is from [16]. Now, restricting $\lambda$ to prevent exponential growth in the number of samples, we get a sample requirement of $n = \Omega(d^2 \log p)$. This matches the lower bound for degree $d$ bounded graphs in [16]. However, note that Theorem 4 is stronger in the sense that the bound holds for a smaller class of graphs *i.e.* only approximately $d$-regular, and not $d$-bounded.

## 5.4 Approximate Edge Bounded Graphs

Let $\mathcal{G}_{p,k}^{\text{approx}}$ be the set of all graphs with number of edges $\in \left[\frac{k}{2}, k\right]$. This class is a subset of the class of graphs with edges at most $k$. Here, we have:

**Theorem 5.** *Consider the class* $\mathcal{G}_{p,k}^{\text{approx}}$, *and let* $k \geq 9$. *If we have number of samples* $n \leq (1 - \delta) \max \left\{ \frac{\log\left(\frac{k}{2}\right)}{\lambda \tanh \lambda}, \frac{e^{\lambda(\sqrt{2k}-1)}}{2\lambda e^\lambda(\sqrt{2k}+1)} \log\left(\frac{k}{2}\right) \right\}$, *then* $p_{max} \geq \delta$.

Note that the second term in the bound above is from [16]. If we restrict $\lambda$ to prevent exponential growth in the number of samples, we get a sample requirement of $n = \Omega(k \log k)$. Again, we match the lower bound for the edge bounded class in [16], but through a smaller class.

# 6 Erdős-Rényi graphs $G(p, c/p)$

In this section, we relate the number of samples required to learn $G \sim G(p, c/p)$ for the dense case, for guaranteeing a constant average probability of error $p_{\text{avg}}$. We have the following main result whose proof can be found in the Appendix.

**Theorem 6.** *Let* $G \sim G(p, c/p)$, $c = \Omega(p^{3/4} + \epsilon')$, $\epsilon' > 0$. *For this class of random graphs, if* $p_{avg} \leq 1/90$, *then* $n \geq \max(n_1, n_2)$ *where:*

$$n_1 = \frac{H(c/p)(3/80)\left(1 - 80 p_{\text{avg}} - O(1/p)\right)}{\left(\frac{4\lambda p}{3} \exp(-\frac{p}{36}) + 4\exp(-\frac{p^{\frac{3}{2}}}{144}) + \frac{4\lambda}{9\left(1+(\cosh(2\lambda))^{\frac{c^2}{6p}}\right)}\right)}, n_2 = \frac{p}{4} H(c/p)(1 - 3 p_{\text{avg}}) - O(1/p)$$

$$(6)$$

**Remark 3.** *In the denominator of the first expression, the dominating term is* $\frac{4\lambda}{9\left(1+(\cosh(2\lambda))^{\frac{c^2}{6p}}\right)}$.

*Therefore, we have the following corollary.*

**Corollary 4.** *Let* $G \sim G(p, c/p)$, $c = \Omega(p^{3/4+\epsilon'})$ *for any* $\epsilon' > 0$. *Let* $p_{\mathrm{avg}} \leq 1/90$, *then*

1. $\lambda = \Omega(\sqrt{p}/c) : \Omega\left(\lambda H(c/p)(\cosh(2\lambda))^{\frac{c^2}{6p}}\right)$ *samples are needed.*

2. $\lambda < O(\sqrt{p}/c) : \Omega(c\log p)$ *samples are needed. (This bound is from [1] )*

**Remark 4.** *This means that when* $\lambda = \Omega(\sqrt{p}/c)$, *a huge number (exponential for constant* $\lambda$*) of samples are required. Hence, for any efficient algorithm, we require* $\lambda = O\left(\sqrt{p}/c\right)$ *and in this regime* $O\left(c\log p\right)$ *samples are required to learn.*

### 6.1 Proof Outline

The proof skeleton is based on Lemma 2. The essence of the proof is to cover a set of graphs $\mathcal{T}$, with large measure, by an exponentially small set where the KL-divergence between any covered and the covering graph is also very small. For this we use Corollary 3. The key steps in the proof are outlined below:

1. We identify a subclass of graphs $\mathcal{T}$, as in Lemma 2, whose measure is close to 1, i.e. $\mu(\mathcal{T}) = 1 - o(1)$. A natural candidate is the 'typical' set $\mathcal{T}_\epsilon^p$ which is defined to be a set of graphs each with $(\frac{cp}{2} - \frac{cp\epsilon}{2}, \frac{cp}{2} + \frac{cp\epsilon}{2})$ edges in the graph.

2. (Path property) We show that most graphs in $\mathcal{T}$ have property $\mathcal{R}$: there are $O(p^2)$ pairs of nodes such that every pair is *well connected* by $O(\frac{c^2}{p})$ node disjoint paths of length 2 with high probability. The measure $\mu(\mathcal{R}\,|\mathcal{T}) = 1 - \delta_1$.

3. (Covering with low diameter) *Every graph $G$ in $\mathcal{R}\bigcap\mathcal{T}$ is covered by a graph $G'$ from a covering set $C_\mathcal{R}(\delta_2)$ such that their edge set differs **only** in the $O(p^2)$ nodes that are well connected.* Therefore, by Corollary 3, KL-divergence between $G$ and $G'$ is very small $(\delta_2 = O(\lambda p^2 \cosh(\lambda)^{-c^2/p}))$.

4. (Efficient covering in Size) Further, the covering set $C_\mathcal{R}$ is exponentially smaller than $\mathcal{T}$.

5. (Uncovered graphs have exponentially low measure) Then we show that the uncovered graphs have large KL-divergence $\left(O(p^2\lambda)\right)$ but their measure $\mu(\mathcal{R}^c\,|\mathcal{T})$ is exponentially small.

6. Using a similar (but more involved) expression for probability of error as in Corollary 2, roughly we need $O(\frac{\log|T|}{\delta_1+\delta_2})$ samples.

The above technique is very general. Potentially this could be applied to other random graph classes.

## 7 Summary

In this paper, we have explored new approaches for computing sample complexity lower bounds for Ising models. By explicitly bringing out the dependence on the weights of the model, we have shown that unless the weights are restricted, the model may be hard to learn. For example, it is hard to learn a graph which has many paths between many pairs of vertices, unless $\lambda$ is controlled. For the random graph setting, $\mathcal{G}_{p,c/p}$, while achievability is possible in the $c = \mathrm{poly}\log p$ case[1], we have shown lower bounds for $c > p^{0.75}$. Closing this gap remains a problem for future consideration. The application of our approaches to other deterministic/random graph classes such as the Chung-Lu model[4] (a generalization of Erdős-Rényi graphs), or small-world graphs[18] would also be interesting.

### Acknowledgments

R.T. and P.R. acknowledge the support of ARO via W911NF-12-1-0390 and NSF via IIS-1149803, IIS-1320894, IIS-1447574, and DMS-1264033. K.S. and A.D. acknowledge the support of NSF via CCF 1422549, 1344364, 1344179 and DARPA STTR and a ARO YIP award.

## Footnotes

[1]Note that a lower bound for a restricted subset of a class of Ising models will also serve as a lower bound for the class without that restriction.

[2]Both the bound from [16] and the bound from Lemma 3 have exponential asymptotic behaviour (*i.e.* as $k$ grows) for constant $\lambda$. For smaller $\lambda$, the bound from [16] is strictly better. However, not all graph classes allow for the presence of a *large enough* clique, for e.g., girth bounded graphs, path restricted graphs, Erdős-Rényi graphs.

[3]In fact, for $\lambda \leq 3$, we have $e^{\lambda^2/2} \leq \cosh(2\lambda) \leq e^{2\lambda^2}$. For $\lambda > 3$, $\cosh(2\lambda) > 200$

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
