[Supplementary Material]

# 8 Appendix A - Proofs for Section 3 and Section 4

## 8.1 Proof of Lemma 1

*Proof.* Starting with the original statement of Fano's lemma (see [5, Theorem 2.10.1]), we get:

$$p_{avg} \geq \frac{H(G) - I(G; \phi(X^n)) - \log 2}{\log|\mathcal{G}|}$$

$$\overset{a}{\geq} \frac{H(G) - I(G; X^n) - \log 2}{\log|\mathcal{G}|} \tag{7}$$

Here we have: (a) by the Data Processing Inequality (see [5, Theorem 2.8.1])

Now, note that:

$$p_{avg} = \sum_{G \in \mathcal{G}} \mathrm{Pr}_\mu(G).\mathrm{Pr}\left(\hat{G} \neq G\right) \leq \max_{G \in \mathcal{G}} \mathrm{Pr}\left(\hat{G} \neq G\right) = p_{\max} \tag{8}$$

$\square$

## 8.2 Proof of Corollary 1

*Proof.* We get the stated bound by picking $\mu$ to be a uniform measure on $\mathcal{G}$ in Lemma 1, and then using: $H(G) = \log|\mathcal{G}|$ and $I(G; X^n) \leq H(X^n) \leq np$. $\square$

## 8.3 Proof of Lemma 2

*Proof.* The conditional version of Fano's lemma (see [1, Lemma 9]) yields:

$$\mathbb{E}_\mu\left[\mathrm{Pr}\left(\hat{G} \neq G\right) \middle| G \in \mathcal{T}\right] \geq \frac{H(G|G \in \mathcal{T}) - I(G; X^n|G \in \mathcal{T}) - \log 2}{\log|\mathcal{T}|} \tag{9}$$

Now,

$$
\begin{aligned}
p_{avg} &= \mathbb{E}_\mu\left[\mathrm{Pr}\left(\hat{G} \neq G\right)\right] \\
&= \mathrm{Pr}_\mu(G \in \mathcal{T})\mathbb{E}_\mu\left[\mathrm{Pr}\left(\hat{G} \neq G\right) \middle| G \in \mathcal{T}\right] + \mathrm{Pr}_\mu(G \notin \mathcal{T})\mathbb{E}_\mu\left[\mathrm{Pr}\left(\hat{G} \neq G\right) \middle| G \notin \mathcal{T}\right] \\
&\overset{a}{\geq} \mathrm{Pr}_\mu(G \in \mathcal{T})\mathbb{E}_\mu\left[\mathrm{Pr}\left(\hat{G} \neq G\right) \middle| G \in \mathcal{T}\right] \\
&\overset{b}{\geq} \mu(\mathcal{T})\frac{H(G|G \in \mathcal{T}) - I(G; X^n|G \in \mathcal{T}) - \log 2}{\log|\mathcal{T}|}
\end{aligned}
\tag{10}
$$

Here we have: (a) since both terms in the equation before are positive. (b) by using the conditional Fano's lemma.

Also, note that:

$$
\begin{aligned}
\mathbb{E}_\mu\left[\mathrm{Pr}\left(\hat{G} \neq G\right) \middle| G \in \mathcal{T}\right] &= \sum_{G \in \mathcal{T}} \mathrm{Pr}_\mu(G|G \in \mathcal{T}).\mathrm{Pr}\left(\hat{G} \neq G\right) \\
&\leq \max_{G \in \mathcal{T}} \mathrm{Pr}\left(\hat{G} \neq G\right) \\
&\leq \max_{G \in \mathcal{G}} \mathrm{Pr}\left(\hat{G} \neq G\right) = p_{\max}
\end{aligned}
\tag{11}
$$

$\square$

## 8.4 Proof of Corollary 2

*Proof.* We pick $\mu$ to be a uniform measure and use $H(G) = \log|\mathcal{G}|$. In addition, we upper bound the mutual information through an approach in [20] which relates it to coverings in terms of the

KL-divergence as follows:

$$
\begin{aligned}
I(G; X^n | G \in \mathcal{T}) &\overset{a}{=} \sum_{G \in \mathcal{T}} P_\mu(G | G \in \mathcal{T}) D\left(f_G(\mathbf{x}^n) \| f_X(\mathbf{x}^n)\right) \\
&\overset{b}{\leq} \sum_{G \in \mathcal{T}} P_\mu(G | G \in \mathcal{T}) D\left(f_G(\mathbf{x}^n) \| Q(\mathbf{x}^n)\right) \\
&\overset{c}{=} \sum_{G \in \mathcal{T}} P_\mu(G | G \in \mathcal{T}) D\left(f_G(\mathbf{x}^n) \Bigg\| \sum_{G' \in C_\mathcal{T}(\epsilon)} \frac{1}{|C_\mathcal{T}(\epsilon)|} f_{G'}(\mathbf{x}^n)\right) \\
&= \sum_{G \in \mathcal{T}} P_\mu(G | G \in \mathcal{T}) \sum_{\mathbf{x}^n} f_G(\mathbf{x}^n) \log\left(\frac{f_G(\mathbf{x}^n)}{\sum_{G' \in C_\mathcal{T}(\epsilon)} \frac{1}{|C_\mathcal{T}(\epsilon)|} f_{G'}(\mathbf{x}^n))}\right) \\
&\overset{d}{\leq} \log|C_\mathcal{T}(\epsilon)| + n\epsilon
\end{aligned}
\tag{12}
$$

Here we have: (a) $f_X(\cdot) = \sum_{G \in \mathcal{T}} P_\mu(G | G \in \mathcal{T}) f_G(\cdot)$ . (b) $Q(\cdot)$ is any distribution on $\{-1, 1\}^{np}$ (see [20, Section 2.1]). (c) by picking $Q(\cdot)$ to be the average of the set of distributions $\{f_G(\cdot), G \in C_\mathcal{T}(\epsilon)\}$. (d) by lower bounding the denominator sum inside the $\log$ by only the covering element term for each $G \in \mathcal{T}$. Also using $D\left(f_G(\mathbf{x}^n) \| f_{G'}(\mathbf{x}^n)\right) = nD\left(f_G \| f'_G\right) (\leq n\epsilon)$, since the samples are drawn i.i.d.

Plugging these estimates in Lemma 2 gives the corollary. $\qquad\square$

## 8.5 Proof of Lemma 3

*Proof.* Consider a graph $G(V, E)$ with two nodes $a$ and $b$ such that there are at least $d$ node disjoint paths of length at most $\ell$ between $a$ and $b$. Consider another graph $G'(V, E')$ with edge set $E' \subseteq E$ such that $E'$ contains only edges belonging to the $d$ node disjoint paths of length $\ell$ between $a$ and $b$. All other edges are absent in $E'$. Let $\mathcal{P}$ denote the set of node disjoint paths. By Griffith's inequality (see [7, Theorem 3.1] ),

$$
\begin{aligned}
\mathbb{E}_{f_G}\left[x_a x_b\right] &\geq \mathbb{E}_{f_{G'}}\left[x_a x_b\right] \\
&= 2P_{G'}\left(x_a x_b = +1\right) - 1
\end{aligned}
\tag{13}
$$

Here, $P_{G'}(.)$ denotes the probability of an event under the distribution $f_{G'}$.

We will calculate the ratio $P_{G'}\left(x_a x_b = +1\right) / P_{G'}\left(x_a x_b = -1\right)$. Since we have a *zero-field* ising model (*i.e.* no weight on the nodes), $f_{G'}(\mathbf{x}) = f_{G'}(-\mathbf{x})$. Therefore, we have:

$$
\frac{P_{G'}\left(x_a x_b = +1\right)}{P_{G'}\left(x_a x_b = -1\right)} = \frac{2P_{G'}\left(x_a = +1, x_b = +1\right)}{2P_{G'}\left(x_a = -1, x_b = +1\right)}
\tag{14}
$$

Now consider a path $p \in \mathcal{P}$ of length $\ell_p$ whose end points are $a$ and $b$. Consider an edge $(i, j)$ in the path $p$. We say $i, j$ disagree if $x_i$ and $x_j$ are of opposite signs. Otherwise, we say they agree. When $x_b = +1$, $x_a$ is $+1$ iff there are even number of disagreements in the path $p$. Odd number of disagreements would correspond to $x_a = -1$, when $x_b = +1$. The location of the disagreements exactly specifies the signs on the remaining variables, when $x_b = +1$. Let $d(p)$ denote the number of disagreements in path $p$. Every agreement contributes a term $\exp(\lambda)$ and every disagreement

contributes a term $\exp(-\lambda)$. Now, we use this to bound (14) as follows:

$$\frac{P_{G'}\left(x_a x_b = +1\right)}{P_{G'}\left(x_a x_b = -1\right)} \overset{a}{=} \frac{\prod\limits_{p \in \mathcal{P}}\left(\sum\limits_{d(p) \text{ even}} e^{\lambda \ell_p} e^{-2\lambda d(p)}\right)}{\prod\limits_{p \in \mathcal{P}}\left(\sum\limits_{d(p) \text{ odd}} e^{\lambda \ell_p} e^{-2\lambda d(p)}\right)}$$

$$\overset{b}{=} \frac{\prod\limits_{p \in \mathcal{P}}\left((1+e^{-2\lambda})^{\ell_p} + (1-e^{-2\lambda})^{\ell_p}\right)}{\prod\limits_{p \in \mathcal{P}}\left((1+e^{-2\lambda})^{\ell_p} - (1-e^{-2\lambda})^{\ell_p}\right)} \quad \overset{c}{=} \frac{\prod\limits_{p \in \mathcal{P}}\left(1 + (\tanh(\lambda))^{\ell_p}\right)}{\prod\limits_{p \in \mathcal{P}}\left(1 - (\tanh(\lambda))^{\ell_p}\right)} \tag{15}$$

$$\overset{d}{\geq} \frac{\left(1 + (\tanh(\lambda))^{\ell}\right)^d}{\left(1 - (\tanh(\lambda))^{\ell}\right)^d} \tag{16}$$

Here we have: (a) by the discussion above regarding even and odd disagreements. Further, the partition function $Z$ (of $f_{G'}$) cancels in the ratio and since the paths are disjoint, the marginal splits as a product of marginals over each path. (b) using the binomial theorem to add up the even and odd terms separately. (c) $\ell_p \leq \ell,\ \forall p \in \mathcal{P}$. (d) there are $d$ paths in $\mathcal{P}$.

Substituting in (13), we get:

$$\mathbb{E}_{f_G}\left[x_a x_b\right] \geq 1 - \frac{2}{1 + \frac{(1+(\tanh(\lambda))^{\ell})^d}{(1-(\tanh(\lambda))^{\ell})^d}}. \tag{17}$$

$\square$

### 8.6 Proof of Corollary 3

*Proof.* From Eq. (4), we get:

$$D\left(f_G \| f_{G'}\right) \leq \sum_{(s,t) \in E - E'} \lambda\left(\mathbb{E}_G\left[x_s x_t\right] - \mathbb{E}_{G'}\left[x_s x_t\right]\right) + \sum_{(s,t) \in E' - E} \lambda\left(\mathbb{E}_{G'}\left[x_s x_t\right] - \mathbb{E}_G\left[x_s x_t\right]\right)$$

$$\overset{a}{\leq} \sum_{(s,t) \in E - E'} \lambda\left(1 - \mathbb{E}_{G'}\left[x_s x_t\right]\right) + \sum_{(s,t) \in E' - E} \lambda\left(1 - \mathbb{E}_G\left[x_s x_t\right]\right)$$

$$\overset{b}{\leq} \frac{2\lambda |E - E'|}{1 + \frac{(1+(\tanh(\lambda))^{\ell})^d}{(1-(\tanh(\lambda))^{\ell})^d}} + \frac{2\lambda |E' - E|}{1 + \frac{(1+(\tanh(\lambda))^{\ell})^d}{(1-(\tanh(\lambda))^{\ell})^d}} \tag{18}$$

Here we have: (a) $\mathbb{E}_G\left[x_s x_t\right] \leq 1$ and $\mathbb{E}_{G'}\left[x_s x_t\right] \leq 1$ (b) for any $(s,t) \in E - E'$, the pair of nodes are $(\ell, d)$ connected. Therefore, bound on $\mathbb{E}_{G'}\left[x_s x_t\right]$ from Lemma 3 applies. Similar bound holds for $\mathbb{E}_G\left[x_s x_t\right]$ for $(s,t) \in E' - E$. $\square$

### 8.7 Proof of Lemma 4

*Proof.* Since the graphs $G(V, E)$ and $G'(V, E')$ differ by only the edge $(a, b) \in E$, we have:

$$\frac{P_G(x_a x_b = +1)}{P_G(x_a x_b = -1)} = e^{2\lambda} \frac{P_{G'}(x_a x_b = +1)}{P_{G'}(x_a x_b = -1)} \tag{19}$$

Here, $P_G(\cdot)$ corresponds to the probability of an event under $f_G$. Let $q = P_{G'}(x_a x_b = +1)$. Now, writing the difference of the correlations,

$$\mathbb{E}_G\left[x_a x_b\right] - \mathbb{E}_{G'}\left[x_a x_b\right] = 2\left(P_G(x_a x_b = +1) - P_{G'}(x_a x_b = -1)\right)$$

$$\overset{a}{=} 2\left(\frac{e^{2\lambda} q}{1 - q + e^{2\lambda} q} - q\right)$$

$$= 2\left(e^{2\lambda} - 1\right)\left(\frac{q - q^2}{1 - q + e^{2\lambda} q}\right) \tag{20}$$

Here we have: (a) by substituting from (19)

Let $h(q) = \left(\frac{q - q^2}{1 - q + e^{2\lambda}q}\right)$. Since we have $\lambda > 0$ *i.e. ferromagnetic* ising model, we know that $q \in [\frac{1}{2}, 1]$. Also, differentiating $h(q)$, we get:

$$h'(q) = \frac{1 - 2q - \left(e^{2\lambda} - 1\right)q^2}{(1 - q + e^{2\lambda}q)^2} \tag{21}$$

It is easy to check that $h'(q) \leq 0$ for $q \in [\frac{1}{2}, 1]$. Thus, $h(q)$ is a decreasing function, and so, substituting $q = 1/2$ in (20),

$$\mathbb{E}_G\left[x_a x_b\right] - \mathbb{E}_{G'}\left[x_a x_b\right] \leq \frac{e^{2\lambda} - 1}{e^{2\lambda} + 1} = \tanh(\lambda) \tag{22}$$

Also, from Eq. (4),

$$D\left(f_G \| f_{G'}\right) \leq \lambda\left(\mathbb{E}_G\left[x_a x_b\right] - \mathbb{E}_{G'}\left[x_a x_b\right]\right) \leq \lambda \tanh(\lambda) \tag{23}$$

$\square$

# 9    Appendix B - Proofs for Section 5

For the proofs in this section, we will be using the estimate of the number of samples presented in Remark 2. To recapitulate, we had the following generic statement:

For any graph class $\mathcal{G}$ and its subset $\mathcal{T} \subset \mathcal{G}$, suppose we can cover $\mathcal{T}$ with a single point (denoted by $G_0$) with KL-radius $\rho$, *i.e.* for any other $G \in \mathcal{T}$, $D\left(f_G \| f_{G_0}\right) \leq \rho$. Now, if

$$n \leq \frac{\log|\mathcal{T}|}{\rho}(1 - \delta) \tag{24}$$

then $p_{max} \geq \delta$. Note that, assuming $\mathcal{T}$ is growing with $p$, we have ignored the lower order term.

So, for each of the graph classes under consideration, we shall show how to construct $G_0$, $\mathcal{T}$ and compute $\rho$.

## 9.1    Proof of Theorem 1

*Proof.* The graph class is $\mathcal{G}_{p,\eta}$, the set of all graphs on $p$ vertices with at most $\eta$ ($\eta = o(p)$) paths between any two vertices.

**Constructing $G_0$:** We consider the following basic building block. Take two vertices $(s, t)$ and connect them. In addition, take $\eta - 1$ more vertices, and connect them to both $s$ and $t$. Now, there are exactly $\eta$ paths between $(s, t)$. There are $(\eta + 1)$ total nodes and $(2\eta - 1)$ total edges.

Now, take $\alpha$ disjoint copies of these blocks. We note that we must have $\alpha(\eta + 1) \leq p$. We choose $\alpha = \left\lfloor \frac{p}{\eta + 1} \right\rfloor \geq \frac{p}{2(\eta + 1)}$ suffices.

**Constructing $\mathcal{T}$ - Ensemble 1:** Starting with $G_0$, we consider the family of graphs $\mathcal{T}$ obtained by removing the main $(s, t)$ edge from one of the blocks. So, we get $\alpha$ different graphs. Let $G_i$, $i \in [\alpha]$, be the graph obtained by removing this edge from the $i^{th}$ block. Then, note that $G_0$ and $G_i$ only differ by a single pair $(s_i, t_i)$, which is $(2, \eta)$ connected in $G_i$. From Corollary 3 we have, $D\left(f_{G_0} \| f_{G_i}\right) \leq \frac{2\lambda}{1 + \cosh(2\lambda)^{\eta - 1}} = \rho$. Plugging $|\mathcal{T}| = \alpha$, and $\rho$ into Eq. (24) gives us the second term for the bound in the theorem.

**Constructing $\mathcal{T}$ - Ensemble 2:** Starting with $G_0$, we consider the family of graphs $\mathcal{T}$ obtained by removing any edge from one of the blocks. So, we get $\alpha(2\eta - 1) \geq \frac{p}{2}$ different graphs. Let $G_i$ be any such graph. Then, note that $G_0$ and $G_i$ only differ by a single edge. From Lemma 4 we have, $D\left(f_{G_0} \| f_{G_i}\right) \leq \lambda \tanh(\lambda) = \rho$. Plugging $|\mathcal{T}| \geq p/2$, and $\rho$ into Eq. (24) gives us the first term for the bound in the theorem. $\square$

## 9.2 Proof of Theorem 2

*Proof.* The graph class is $\mathcal{G}_{p,\eta,\gamma}$, the set of all graphs on $p$ vertices with at most $\eta$ paths of length at most $\gamma$ between any two vertices.

**Constructing** $G_0$: We consider the following basic building block. Take two vertices $(s,t)$ and connect them. In addition, take $\eta - 1$ more vertices, and connect them to both $s$ and $t$. Also, take another $k$ vertex disjoint paths, each of length $\gamma + 1$, between $(s,t)$. Now, there are exactly $\eta + k$ paths between $(s,t)$, but at most $\eta$ paths of length at most $\gamma$. There are $(k\gamma + \eta + 1)$ total nodes and $(k(\gamma + 1) + 2\eta - 1)$ total edges.

Now, take $\alpha$ disjoint copies of these blocks. Note that we must choose $\alpha$ and $k$ such that $\alpha(k\gamma + \eta + 1) \leq p$. For some $\nu \in (0,1)$, we choose $\alpha = p^\nu$. In this case, $k = t_\nu = \frac{p^{1-\nu} - (\eta + 1)}{\gamma}$ suffices.

**Constructing** $\mathcal{T}$ **- Ensemble 1**: Starting with $G_0$, we consider the family of graphs $\mathcal{T}$ obtained by removing the main $(s,t)$ edge from one of the blocks. So, we get $\alpha$ different graphs. Let $G_i$, $i \in [\alpha]$, be the graph obtained by removing this edge from the $i^{th}$ block. Then, note that $G_0$ and $G_i$ only differ by a single pair $(s_i, t_i)$, which is $(2, \eta - 1)$ connected and also $(t_\nu, \gamma + 1)$ connected, in $G_i$. Based on the proof of Lemma 3, the estimate of $D\left(f_{G_i} \| f_{G_0}\right)$ can be recomputed by handling the two different sets of correlation contributions from the two sets of node disjoint paths, and then combining them based on the probabilities. We get, $D\left(f_{G_0} \| f_{G_i}\right) \leq \dfrac{2\lambda}{1 + \left[\cosh(2\lambda)^{\eta-1}\left(\frac{1+\tanh(\lambda)^{\gamma+1}}{1-\tanh(\lambda)^{\gamma+1}}\right)^{t_\nu}\right]} = \rho.$

Plugging $|\mathcal{T}| = \alpha$, and $\rho$ into Eq. (24) gives us the second term for the bound in the theorem.

**Constructing** $\mathcal{T}$ **- Ensemble 2**: Starting with $G_0$, we consider the family of graphs $\mathcal{T}$ obtained by removing any edge from one of the blocks. So, we get $\alpha(k(\gamma + 1) + 2\eta - 1) \geq \frac{p}{2}$ different graphs. Let $G_i$ be any such graph. Then, note that $G_0$ and $G_i$ only differ by a single edge. From Lemma 4 we have, $D\left(f_{G_0} \| f_{G_i}\right) \leq \lambda \tanh(\lambda) = \rho$. Plugging $|\mathcal{T}|$ and $\rho$ into Eq. (24) gives us the second term for the bound in the theorem. □

## 9.3 Proof of Theorem 3

*Proof.* The graph class is $\mathcal{G}_{p,g,d}$, the set of all graphs on $p$ vertices with girth atleast $g$ and degree at most $d$.

**Constructing** $G_0$: We consider the following basic building block. Take two vertices $(s,t)$ and connect them. In addition, take $k$ vertex disjoint paths, each of length $g - 1$ between $(s,t)$. Now, there are exactly $k$ paths between $(s,t)$. There are $(k(g-2) + 2)$ total nodes and $(k(g-1) + 1)$ total edges.

Now, take $\alpha$ disjoint copies of these blocks. Note that we must choose $\alpha$ and $k$ such that $\alpha(k(g-2) + 2) \leq p$. For some $\nu \in (0,1)$, we choose $\alpha = p^\nu$. In this case, $k = d_\nu = \min\left(d, \frac{p^{1-\nu}}{g}\right)$ suffices.

**Constructing** $\mathcal{T}$ **- Ensemble 1**: Starting with $G_0$, we consider the family of graphs $\mathcal{T}$ obtained by removing the main $(s,t)$ edge from one of the blocks. So, we get $\alpha$ different graphs. Let $G_i$, $i \in [\alpha]$, be the graph obtained by removing this edge from the $i^{th}$ block. Then, note that $G_0$ and $G_i$ only differ by a single pair $(s_i, t_i)$, which is $(d_\nu, g - 1)$ connected in $G_i$. From Corollary 3 we have, $D\left(f_{G_0} \| f_{G_i}\right) \leq \dfrac{2\lambda}{1 + \left(\frac{1+\tanh(\lambda)^{g-1}}{1-\tanh(\lambda)^{g-1}}\right)^{d_\nu}} = \rho$. Plugging $|\mathcal{T}| = \alpha$, and $\rho$ into Eq. (24) gives us the second term for the bound in the theorem.

**Constructing** $\mathcal{T}$ **- Ensemble 2**: Starting with $G_0$, we consider the family of graphs $\mathcal{T}$ obtained by removing any edge from one of the blocks. So, we get $\alpha(k(g-1) + 1) \geq \frac{p}{2}$ different graphs. Let $G_i$ be any such graph. Then, note that $G_0$ and $G_i$ only differ by a single edge. From Lemma 4 we have, $D\left(f_{G_0} \| f_{G_i}\right) \leq \lambda \tanh(\lambda) = \rho$. Plugging $|\mathcal{T}|$ and $\rho$ into Eq. (24) gives us the second term for the bound in the theorem. □

### 9.4 Proof of Theorem 4

*Proof.* The graph class is $\mathcal{G}_{p,d}^{\text{approx}}$, the set of all graphs on $p$ vertices with degree either $d$ or $d-1$ (we assume that $p$ is a multiple of $d+1$ - if not, we can instead look at a smaller class by ignoring at most $d$ vertices). The construction here is the same as in [16].

**Constructing $G_0$:** We divide the vertices into $p/(d+1)$ groups, each of size $d+1$, and then form cliques in each group.

**Constructing $\mathcal{T}$:** Starting with $G_0$, we consider the family of graphs $\mathcal{T}$ obtained by removing any one edge. Thus, we get $\frac{p}{d+1}\binom{d+1}{2} \geq \frac{pd}{4}$ such graphs. Also, any such graph, $G_i$, differs from $G_0$ by a single edge, and also, differs only in a pair that is part of a clique minus one edge. So, combining the estimates from [16] and Lemma 4, we have, $D\left(f_{G_0}\|f_{G_i}\right) \leq \min\left(\frac{2\lambda d e^\lambda}{e^{\lambda d}}, \lambda\tanh(\lambda)\right) = \rho$. Plugging $|\mathcal{T}|$ and $\rho$ into Eq. (24) gives us the theorem. $\qquad\square$

### 9.5 Proof of Theorem 5

*Proof.* The graph class is $\mathcal{G}_{p,k}^{\text{approx}}$, the set of all graphs on $p$ vertices with at most $k$ edges. The construction here is the same as in [16]

**Constructing $G_0$:** We choose a largest possible number of vertices $m$ such that we can have a clique on them *i.e.* $\binom{m}{2} \leq k$. Then, $\sqrt{2k}+1 \geq m \geq \sqrt{2k}-1$. We ignore any unused vertices.

**Constructing $\mathcal{T}$:** Starting with $G_0$, we consider the family of graphs $\mathcal{T}$ obtained by removing any one edge. Thus, we get $\binom{m}{2} \geq \frac{k}{2}$ such graphs. Also, any such graph, $G_i$, differs from $G_0$ by a single edge, and also, differs only in a pair that is part of a clique minus one edge. So, combining the estimates from [16] and Lemma 4, we have, $D\left(f_{G_0}\|f_{G_i}\right) \leq \min\left(\frac{2\lambda e^\lambda(\sqrt{2k}+1)}{e^{\lambda(\sqrt{2k}-1)}}, \lambda\tanh(\lambda)\right) = \rho$. Plugging $|\mathcal{T}|$ and $\rho$ into Eq. (24) gives us the theorem. $\qquad\square$

## 10 Appendix C: Proof of Theorem 6

In this section, we outline the covering arguments in detail along with a Fano's Lemma variant to prove Theorem 6.

We recall some definitions and results from [1].

**Definition 3.** *Let $\mathcal{T}_\epsilon^n = \{G : |\bar{d}(G) - c| \leq c\epsilon\}$ denote the $\epsilon$-typical set of graphs where $\bar{d}(G)$ is the ratio of sum of degree of nodes to the total number of nodes.*

A graph $G$ on $p$ nodes is drawn according to the distribution characterizing the Erdős-Rényi ensemble $G(p, c/p)$ (also denoted $\mathcal{G}_{\text{ER}}$ without the parameter $c$). Then $n$ i.i.d samples $\mathbf{X}^n = \mathbf{X}^{(1)}, \ldots \mathbf{X}^{(n)}$ are drawn according to $f_G(\mathbf{x})$ with the scalar weight $\lambda > 0$. Let $H(\cdot)$ denote the binary entropy function.

**Lemma 5.** *(Lemma $8, 9$ and Proof of Theorem $4$ in [1] ) The $\epsilon$- typical set satisfies:*

1. *$P_{G\sim G(p,c/p)}\left(G \in \mathcal{T}_\epsilon^p\right) = 1 - a_p$ where $a_p \to 0$ as $p \to \infty$.*

2. *$2^{-\binom{p}{2}H(c/p)(1+\epsilon)} \leq P_{G\sim G(p,c/p)}\left(G\right) \leq 2^{-\binom{p}{2}H(c/p)}$.*

3. *$(1-\epsilon)2^{\binom{p}{2}H(c/p)} \leq |\mathcal{T}_\epsilon^p| \leq 2^{\binom{p}{2}H(c/p)(1+\epsilon)}$ for sufficiently large $p$.*

4. *$H(G|G \in \mathcal{T}_\epsilon^p) \geq \binom{p}{2}H(c/p)$.*

5. *(Conditional Fano's Inequality:)*

$$P(\hat{G}(\mathbf{X}^n) \neq G | G \in \mathcal{T}_\epsilon^p) \geq \frac{H(G|G \in \mathcal{T}_\epsilon^p) - I(G; \mathbf{X}^n | G \in \mathcal{T}_\epsilon^p) - 1}{\log_2 |\mathcal{T}_\epsilon^p|} \qquad (25)$$

## 10.1 Covering Argument through Fano's Inequality

Now, we consider the random graph class $G(p, c/p)$. Consider a learning algorithm $\phi$. Given a graph $G \sim G(p, c/p)$, and $n$ samples $\mathbf{X}^n$ drawn according to distribution $f_G(\mathbf{x})$ (with weight $\lambda > 0$), let $\hat{G} = \phi(\mathbf{X}^n)$ be the output of the learning algorithm. Let $f_X(.)$ be the marginal distribution of $\mathbf{X}^n$ sampled as described above. Then the following holds for $p_{\text{avg}}$ :

$$
\begin{aligned}
p_{\text{avg}} &= \mathbb{E}_{G(p,c/p)} \left[ \mathbb{E}_{\mathbf{X}^n \sim f_G} \left[ \mathbf{1}_{\hat{G} \neq G} \right] \right] \\
&\geq \Pr_{G(p,c/p)} (G \in \mathcal{T}_\epsilon^p) \mathbb{E} \left[ \mathbb{E}_{\mathbf{X}^n \sim f_G} \left[ \mathbf{1}_{\hat{G} \neq G} \right] | G \in \mathcal{T}_\epsilon^p \right] \\
&\overset{a}{=} (1 - a_p) \mathbb{E} \left[ \mathbb{E}_{\mathbf{X}^n \sim f_G} \left[ \mathbf{1}_{\hat{G} \neq G} \right] | G \in \mathcal{T}_\epsilon^p \right] \\
&= (1 - a_p) p'_{avg}
\end{aligned}
\tag{26}
$$

Here, (a) is due to Lemma 5. Here, $p'_{avg}$ is the average probability of error under the conditional distribution obtained by conditioning $G(p, c/p)$ on the event $G \in \mathcal{T}_\epsilon^p$.

Now, consider $G$ sampled according to the conditional distribution $G(p, c/p)|G \in \mathcal{T}_\epsilon^p$. Then, $n$ samples $\mathbf{X}^n$ are drawn i.i.d according to $f_G(\mathbf{x})$. $\hat{G} = \phi(\mathbf{x}^n)$ is the output of the learning algorithm. Applying conditional Fano's inequality from (25) and using estimates from Lemma 5, we have:

$$
\begin{aligned}
p'_{\text{avg}} &= P_{G \sim G(p,c/p)|G \in \mathcal{T}_\epsilon^p, \mathbf{X^n} \sim f_G(\mathbf{x})} \left( \hat{G} \neq G \right) \\
&\overset{a}{\geq} \frac{\binom{p}{2} H(c/p) - I(G; \mathbf{X}^n | G \in \mathcal{T}_\epsilon^p) - 1}{\log_2 |\mathcal{T}_\epsilon^p|} \\
&\overset{b}{\geq} \frac{\binom{p}{2} H(c/p) - I(G; \mathbf{X}^n | G \in \mathcal{T}_\epsilon^p) - 1}{\binom{p}{2} H(c/p)(1 + \epsilon)} \\
&= \frac{1}{1 + \epsilon} - \frac{I(G; \mathbf{X}^n | G \in \mathcal{T}_\epsilon^p)}{\binom{p}{2} H(c/p)(1 + \epsilon)} - \frac{1}{\binom{p}{2} H(c/p)(1 + \epsilon)}
\end{aligned}
\tag{27}
$$

Now, we upper bound $I(G; \mathbf{X}^n | G \in \mathcal{T}_\epsilon^p)$. Now, use a result by Yang and Barron [20] to bound this term.

$$
\begin{aligned}
I(G; \mathbf{X}^n | G \in \mathcal{T}_\epsilon^p) &= \sum_G P_{G(p,c/p)|G \in \mathcal{T}_\epsilon^p}(G) D\left( f_G(\mathbf{x}^n) \| f_X(\mathbf{x}^n) \right) \\
&\leq \sum_G P_{G(p,c/p)|G \in \mathcal{T}_\epsilon^p}(G) D\left( f_G(\mathbf{x}^n) \| Q(\mathbf{x}^n) \right)
\end{aligned}
\tag{28}
$$

where $Q(\cdot)$ is any distribution on $\{-1, 1\}^{np}$. Now, we choose this distribution to be the average of $\{f_G(.), G \in S\}$ where the set $S \subseteq \mathcal{T}_\epsilon^p$ is a set of graphs that is used to 'cover' all the graphs in $\mathcal{T}_\epsilon^p$. Now, we describe the set $S$ together with the covering rules when $c = \Omega(p^{3/4} + \epsilon')$, $\epsilon' > 0$.

## 10.2 The covering set $S$: dense case

First, we discuss certain properties that most graphs in $\mathcal{T}_\epsilon^p$ possess building on Lemma 3. Using these properties, we describe the covering set $S$.

Consider a graph $G$ on $p$ nodes. Divide the node set into three equal parts $A$, $B$ and $C$ of equal size $(p/3)$. Two nodes $a \in A$ and $c \in C$ are $(2, \gamma)$ connected through $B$ if there are at least $\gamma$ nodes in $B$ which are connected to both $a$ and $c$ (with parameter $\gamma$ as defined in Section 4.3). Let $\mathcal{D}(G) \subseteq A \times C$ be the set of all pairs $(a, c)$ : $a \in A$, $c \in C$ such that nodes $a$ and $c$ are $(2, \gamma)$ connected. Let $|\mathcal{D}(G)| = m_{A,C}$. Let $E(G)$ denote the edge in graph $G$.

### 10.2.1 Technical results on $\mathcal{D}(G)$

Nodes $a \in A$ and $c \in C$ are clearly $(2, d)$-connected if there are $d$ nodes in $B$ which are connected to both $a$ and $b$ as it will mean $d$ disjoint paths connecting $a$ and $b$ through the partition $B$. Now if $G \sim G(p, c/p)$, then expected number of disjoint paths between $a$ and $c$ through $B$ is $\frac{p}{3} \frac{c^2}{p^2}$ since

the probability of a path existing through a node $b \in B$ is $\frac{c^2}{p^2}$. Let $n_{a,c}$ be the number of such paths between $a$ and $c$. The event that there is a path through $b_1 \in B$ is independent of the event that there is a path through $b_2 \in B$, applying chernoff bounds (see [12]) for $p/3$ independent bernoulli variables we have:

**Lemma 6.** $\Pr\left(n_{a,c} \leq \frac{c^2}{3p} - \sqrt{4p\log p}\right) \leq \frac{1}{p^2}$ *for any two nodes* $a \in A$ *and* $c \in C$ *when* $G \sim$ $G(p, c/p)$. *The bound is useful for* $c = \Omega(p^{\frac{3}{4}+\epsilon'})$, $\epsilon' > 0$.

Therefore, in this regime of dense graphs, any two nodes in partitions $A$ and $C$ are $(2, \gamma = c^2/6p)$ connected with probability $1 - \frac{1}{p^2}$.

Given $a \in A$ and $c \in C$, the probability that $a$ and $c$ are $(2, \gamma)$ connected is $1 - \frac{1}{p^2}$. The expected number of pairs in $A \times C$ that are $(2, \gamma)$ connected is $(p/3)^2 \left(1 - \frac{1}{p^2}\right)$. Let $\mathcal{D}(G) \subseteq A \times C$ be the set of all pairs $(a, c) : a \in A, c \in C$ such that nodes $a$ and $c$ are $(2, \gamma)$ connected. Let $m_{A,C} = |\mathcal{D}|$. Then we have the following concentration result on $m_{A,C}$:

**Lemma 7.** $\Pr\left(m_{A,C} \leq \frac{1}{2}(p/3)^2\right) \leq b_p = p/3 \exp(-(p/36))$ *when* $G \sim G(p, c/p)$, $c = \Omega(p^{\frac{3}{4}+\epsilon'})$, $\epsilon' > 0$.

*Proof.* The event that the pair $(a_1, c_1) \in A \times C$ is $(2, \gamma)$ connected and the event that the pair $(a_2, c_2) \in A \times C$ are dependent if $a_1 = a_2$ or $c_1 = c_2$. Therefore, we need to obtain a concentration result for the case when you have $(p/3)^2$ Bernoulli variables (each corresponding to a pair in $A \times C$ being $(2, \gamma)$ connected ) which are dependent.

Consider a complete bipartite graph between $A$ and $C$. Since, $|A| = |C| = p/3$. Edges of every complete bipartite graph $K_{p/3,p/3}$ can be decomposed into a disjoint union of $p/3$ perfect matchings between the partitions (this is due to Hall's Theorem repeatedly applied on graphs obtained by removing perfect matchings. See [9] ). Therefore, the set of pairs $A \times C = \bigcup_{1=1}^{p/3} \mathcal{M}_i$ where $\mathcal{M}_i = \{(a_{i_1}, c_{i_1}), \ldots (a_{i_{p/3}}, c_{i_{p/3}})\}$ where all for any $j \neq k$, $a_{i_k} \neq a_{i_j}$ and $c_{i_k} \neq c_{i_j}$.

Let us focus on the number of pairs which are $(2, \gamma)$ connected between $A$ and $C$ in a random graph $G \sim G(p, c/p)$. If $m_{A,C} \leq \frac{1}{2}(p/3)^2$, then at least for one $i$, the number of pairs in $G$ among the pairs in $\mathcal{M}_i$ that are $(2, \gamma)$ is at most $\frac{1}{2}(p/3)$. This is because $(p/3)^2 = \sum_i |\mathcal{M}_i|$. Let $E_i^c$ denote the event that number of edges in $G$ among pairs in $\mathcal{M}_i$ is at most $\frac{1}{2}(p/3)$.

$$\Pr\left(m_{A,C} \leq \frac{1}{2}(p/3)^2\right) \leq \Pr\left(\bigcup_i E_i^c\right) \leq \sum_i \Pr\left(E_i^c\right). \tag{29}$$

The last inequality is due to union bound. A pair in $\mathcal{M}_i$ being $(2, \gamma)$ connected happens with probability $1 - 1/p^2$ from Lemma 6. Since it is a perfect matching, all these events are independent. Let $c_G(\mathcal{M}_i)$ be the number of pairs in $\mathcal{M}_i$ which are $(2, \gamma)$ connected. Therefore, applying a chernoff bound (see [12] Theorem 18.22) for independent Bernoulli variables, we have:

$$\begin{aligned}
\Pr(E_i^c) &= \Pr\left(c_G(\mathcal{M}_i) \leq \mathbb{E}[(p/3)(1/2)\right) \\
&= \Pr\left(c_G(\mathcal{M}_i) \leq \mathbb{E}[c_G(\mathcal{M}_i)] - (p/3)(1/2 - 1/p^2)\right) \\
&\overset{\text{(chernoff)}}{\leq} \exp\left(-(p/3)^2(1/2 - 1/p^2)^2/2(p/3)\right) \\
&\overset{a}{\leq} \exp\left(-(p/36)\right)
\end{aligned}$$

(a) holds for large $p$, i.e. for $p \geq p_0$ such that $(1/2 - 1/p_o^2)^2 \geq 1/6$. Simple calculation shows that $p_0$ can be taken to be greater than or equal to 10.

Now, applying this to (29), we have $\forall p \geq 10$:

$$\Pr\left(m_{A,C} \leq \frac{1}{2}(p/3)^2\right) \leq b_p = p/3 \exp(-(p/36)). \tag{30}$$

$\square$

Let $E(G)$ be the set of edges in $G$.

**Lemma 8.** $Pr\left(\left|\frac{||E(G)\bigcap\mathcal{D}(G)|}{|\mathcal{D}(G)|} - \frac{c}{p}\right| \geq \frac{c}{p}\epsilon \;\middle|\; m_{A,C} \geq \frac{1}{2}(p/3)^2\right) \leq 2\exp(-\frac{c^2\epsilon^2}{36}) = r_c$ *when* $G \sim$ $G(p,c/p)$, $c = \Omega(p^{\frac{3}{4}+\epsilon'})$, $\epsilon' > 0$.

*Proof.* The presence of an edge between a pair on nodes in $A \times C$ is independent of the value of $m_{A,C}$ or whether the pair belongs to $\mathcal{D}$. This is because a pair of nodes being $(2,\gamma)$ connected depends on the rest of the graph and not on the edges in $\mathcal{D}(G)$. Given $|\mathcal{D}| \geq \frac{1}{2}(p/3)^2$, $|\mathcal{E}(\mathcal{G})\bigcap\mathcal{D}(G)| = \sum_{(i,j)\in\mathcal{D}} \mathbf{1}_{(i,j)\in E(G)}$ is the sum of least $\frac{1}{2}(p/3)^2$ bernoulli variables each with success probability $c/p$. Therefore, applying chernoff bounds we have:

$$Pr\left(\left|\frac{||E(G)\bigcap\mathcal{D}(G)|}{|\mathcal{D}(G)|} - \frac{c}{p}\right| \leq \frac{c}{p}\epsilon \;\middle|\; m_{A,C} \geq \frac{1}{2}(p/3)^2\right) \leq 2\exp\left(-\frac{c^2\epsilon^2}{p^2 2|\mathcal{D}|}|\mathcal{D}|^2\right)$$

$$\overset{a}{\leq} 2\exp\left(-\frac{c^2\epsilon^2}{4}(1/3)^2\right) \qquad (31)$$

$$(32)$$

(a)- This is because $|\mathcal{D}| \geq \frac{1}{2}(p/3)^2$. $\qquad\square$

**Lemma 9.** $Pr\left(\left(\left|\frac{||E(G)\bigcap\mathcal{D}(G)|}{|\mathcal{D}(G)|} - \frac{c}{p}\right| \geq \frac{c}{p}\epsilon\right)\bigcup\left(m_{A,C} \leq \frac{1}{2}(p/3)^2\right)\right) \leq b_p + r_c$, $c = \Omega(p^{\frac{3}{4}+\epsilon'})$, $\epsilon' > 0$.

*Proof.*

$$Pr\left(\left(\left|\frac{||E(G)\bigcap\mathcal{D}(G)|}{|\mathcal{D}(G)|} - \frac{c}{p}\right| \geq \frac{c}{p}\epsilon\right)\bigcup\left(m_{A,C} \leq \frac{1}{2}(p/3)^2\right)\right) \overset{a}{\leq} Pr\left(m_{A,C} \leq \frac{1}{2}(p/3)^2\right)$$

$$+Pr\left(\left|\frac{||E(G)\bigcap\mathcal{D}(G)|}{|\mathcal{D}(G)|} - \frac{c}{p}\right| \geq \frac{c}{p}\epsilon \;\middle|\; m_{A,C} \geq \frac{1}{2}(p/3)^2\right) \leq b_p + r_c$$

$$(33)$$

(a)- is because $\Pr(A\bigcup B) \leq \Pr(A) + \Pr(A^c)\Pr(B|A^c) \leq \Pr(A) + \Pr(B|A^c)$. $\qquad\square$

### 10.2.2 Covering set $S$ and its properties

For any graph $G$, let $G_{\mathcal{D}=\emptyset}$ be the graph obtained by removing any edge (if present) between the pairs of nodes in $\mathcal{D}(G)$. Let $\mathcal{V}$ be the set of graphs on $p$ nodes such that $|\mathcal{D}| = m_{A,C} \geq \frac{1}{2}(p/3)^2$ and $\left|\frac{|E(G)\bigcap\mathcal{D}(G)|}{|\mathcal{D}(G)|} - \frac{c}{p}\right| \leq \frac{c\epsilon}{p}$. Define $\mathcal{R}_\epsilon^p = \mathcal{T}_\epsilon^p\bigcap\mathcal{V}$ to be the set of graphs that are in the $\epsilon$ typical set and also belongs to $\mathcal{V}$. We have seen high probability estimates on $m_{A,C}$ when $G \sim G(p,c/p)$. Now, we state an estimate for $\Pr(\mathcal{R}_\epsilon^p)$ when $G \sim G(p,c/p)|\mathcal{T}_\epsilon^p$.

**Lemma 10.** $\Pr_{G(p,c/p)}\left((\mathcal{R}_\epsilon^p)^c \,|\, G \in \mathcal{T}_\epsilon^p\right) \leq \frac{b_p+r_c}{1-a_p} \leq 2(b_p+r_c)$ *for large* $p$, $c = \Omega(p^{\frac{3}{4}+\epsilon'})$, $\epsilon' > 0$.

*Proof.* Expanding the probability expression in Lemma 9 through conditioning on the events $G \in \mathcal{T}_\epsilon^p$ and $G \in (\mathcal{T}_\epsilon^p)^c$, we have:

$$\Pr\left(\left(\left|\frac{|E(G)\bigcap\mathcal{D}(G)|}{|\mathcal{D}(G)|} - \frac{c}{p}\right| \geq \frac{c}{p}\epsilon\right)\bigcup\left(m_{A,C} \leq \frac{1}{2}(p/3)^2\right)|G \in \mathcal{T}_\epsilon^p\right)\Pr(G \in \mathcal{T}_\epsilon^p) \leq b_p + r_c$$

$$(34)$$

This implies:

$$\Pr\left(\left(\left|\frac{||E(G)\bigcap\mathcal{D}(G)|}{|\mathcal{D}(G)|} - \frac{c}{p}\right| \geq \frac{c}{p}\epsilon\right)\bigcup\left(m_{A,C} \leq \frac{1}{2}(p/3)^2\right)|G \in \mathcal{T}_\epsilon^p\right) \overset{a}{\leq} \frac{b_p+r_c}{1-a_p}$$

$$\overset{b}{\leq} 2(b_p+r_c) \qquad (35)$$

(a) is because of estimate 1 in Lemma 5. (b)- For large $p$, $a_p$ can be made smaller than $1/2$. $\qquad\square$

**Lemma 11.** *[8](Size of a Typical set) For any* $0 \leq p \leq 1$, $m \in \mathbb{Z}^+$ *and a small* $\epsilon > 0$, *let* $\mathcal{N}_\epsilon^{m,p} = \{\mathbf{x} \in \{0,1\}^m : \left| \frac{|\{i : x_i = 1\}|}{m} - p \right| \leq p\epsilon\}$. *Then,* $|\mathcal{N}_\epsilon^{m,p}| = \sum\limits_{mp(1-\epsilon) \leq q \leq mp(1+\epsilon)} \binom{m}{q}$. *Further,* $|\mathcal{N}_\epsilon^{m,p}| \geq (1-\epsilon)2^{mH(p)(1-\epsilon)}$.

**Definition 4.** *(Covering set)* $S = \{G_{\mathcal{D}=\emptyset} | G \in \mathcal{R}_\epsilon^p\}$.

Now, we describe the covering rule for the set $\mathcal{R}_\epsilon^p$. For any $G \in \mathcal{R}_\epsilon^p$, we cover $G$ by $G_{\mathcal{D}=\emptyset}$. Note that, given $G$, by definition, $G_{\mathcal{D}=\emptyset}$ is unique. Therefore, there is no ambiguity and no necessity to break ties. Since, the set $\mathcal{D}(G)$ is dependent only on the edges outside the set of pairs $A \times C$, $\mathcal{D}(G_{\mathcal{D}=\emptyset}) = \mathrm{D(G)}$. Therefore, from a given $G' \in \mathcal{R}_\epsilon^p$, by adding different sets of edges in $\mathcal{D}(G')$, it is possible to obtain elements in $\mathcal{R}^p$ covered by $G'$. We now estimate the size of the covering set $S$ relative to the size of $\mathcal{T}_\epsilon^p$. We show that it is small.

**Lemma 12.** $\frac{\log|S|}{\log|\mathcal{T}_\epsilon^p|} \leq \frac{9}{10}\left(\frac{1+\frac{11}{9}\epsilon}{1+\epsilon}\right) - O(1/p)$ *for large* $p$.

*Proof.* By definition of $\mathcal{R}_\epsilon^p$, for every $G \in \mathcal{R}_\epsilon^p$, $|\mathcal{D}| \geq \frac{1}{2}(p/3)^2$ and the number of edges is in $\mathcal{D}$ is at least $\frac{1}{2}(p/3)^2(c/p)(1+\epsilon)$. And the graph that covers $G$ is $G_{\mathcal{D}=\emptyset}$ where all edges from $\mathcal{D}$ are removed if present in $G$. Let any set of $q$ edges be added to $G_{\mathcal{D}=\emptyset}$ among the pairs of nodes in $\mathcal{D}$ to form $G'$ such that $|\mathcal{D}|(c/p)(1-\epsilon) \leq q \leq |\mathcal{D}|(c/p)(1+\epsilon)$. Then, any such $G'$ belongs to $\mathcal{R}_\epsilon^p$. This follows from the definition of the $\mathcal{R}_\epsilon^p$. And $G'$ is still uniquely covered by $G_{\mathcal{D}=\emptyset}$. Uniqueness follows from the discussion that precedes this Lemma. For every covering graph $G_c \in S$, there are at least $\sum\limits_{|\mathcal{D}(G_c)|(c/p)(1-\epsilon) \leq q \leq |\mathcal{D}(G_c)|(c/p)(1+\epsilon)} \binom{|\mathcal{D}(G_c)|}{q}$ distinct graphs $G \in \mathcal{R}_\epsilon^p$ uniquely covered by $G_c$. Using these observations, we upper bound $|S|$ as follows:

$$
\begin{aligned}
\log|\mathcal{T}_\epsilon^p| &\geq \log\left(\sum_{G_c \in S}|\{G \in \mathcal{R}_\epsilon^p : G \text{ is covered by } G_c\}|\right) \\
&\geq \log\left(\sum_{G_c \in S}\sum_{|\mathcal{D}(G_c)|(c/p)(1-\epsilon) \leq q \leq |\mathcal{D}(G_c)|(c/p)(1+\epsilon)}\binom{|\mathcal{D}(G_c)|}{q}\right) \\
&\stackrel{a}{\geq} \log\left(\sum_{G_c \in S}|\mathcal{N}_\epsilon^{|\mathcal{D}(G_c)|,(c/p)}|\right) \\
&\stackrel{a}{\geq} \log|S| + \log\left((1-\epsilon)2^{\frac{1}{2}(p/3)^2 H(c/p)(1-\epsilon)}\right)
\end{aligned} \tag{36}
$$

(a)- This is due to Lemma 11 and the fact that $|\mathcal{D}| \geq \frac{1}{2}(p/3)^2$. Using (36), we have the following chain of inequalities:

$$
\begin{aligned}
\frac{\log|S|}{\log|\mathcal{T}_\epsilon^p|} &\stackrel{a}{\leq} 1 - \frac{\log(1-\epsilon) + \frac{1}{2}(1-\epsilon)(p/3)^2 H(c/p)}{\binom{p}{2}H(c/p)(1+\epsilon)} \\
&= 1 - O(1/p) - \frac{(1-\epsilon)}{9(1+\epsilon)}(p/p - 1) \\
&\stackrel{b}{\leq} 1 - O(1/p) - \frac{(1-\epsilon)}{10(1+\epsilon)} \\
&= \frac{9}{10}\left(\frac{1+\frac{11}{9}\epsilon}{1+\epsilon}\right) - O(1/p)
\end{aligned} \tag{37}
$$

(a)- Upper bound is used from Lemma 5. (b)- This is valid for $p \geq 10$. $\qquad\square$

### 10.3   Completing the covering argument:dense case

We now resume the covering argument from Section 10.1. Having specified the covering set $S$, let the distribution $Q(\mathbf{x}^n) = \frac{1}{|S|}\sum\limits_{G \in S}f_G(\mathbf{x}^n)$. Let $G_1 \in S$ be some arbitrary graph. Recalling the

upper bound on $I(G; \mathbf{X}^n | G \in \mathcal{T}_\epsilon^p)$ from (28), we have:

$$I(G; \mathbf{X}^n | G \in \mathcal{T}_\epsilon^p) \leq \sum_{G \in \mathcal{T}_\epsilon^p} P_{G(p,c/p)|G \in \mathcal{T}_\epsilon^p}(G) D\left(f_G(\mathbf{x}^n) \| Q(\mathbf{x}^n)\right)$$

$$= \sum_{G \in \mathcal{T}_\epsilon^p} P_{G(p,c/p)|G \in \mathcal{T}_\epsilon^p}(G) \sum_{\mathbf{x}^n} f_G(\mathbf{x}^n) \log \frac{f_G(\mathbf{x}^n}{\frac{1}{|S|} \sum_{G \in S} f_G(\mathbf{x}^n)}$$

$$\leq \log|S| + \sum_{G \in (\mathcal{R}_\epsilon^p)^c} P_{G(p,c/p)|G \in \mathcal{T}_\epsilon^p}(G) D\left(f_G(\mathbf{x}^n) \| f_{G_1}(\mathbf{x}^n)\right)$$

$$+ \sum_{G \in \mathcal{R}_\epsilon^p} P_{G(p,c/p)|G \in \mathcal{T}_\epsilon^p}(G) D\left(f_G(\mathbf{x}^n) \| f_{G_{\mathcal{D}=\emptyset}}(\mathbf{x}^n)\right)$$

$$\overset{a}{\leq} \log|S| + 2n\lambda\binom{p}{2}(2b_p + 2r_c) + n2\lambda(p/3)^2 \frac{1}{1 + \frac{(1+(\tanh(\lambda))^2)^\gamma}{(1-(\tanh(\lambda))^2)^\gamma}} \quad (38)$$

$$\leq \log|S| + n\left(2\lambda\binom{p}{2}(2b_p + 2r_c) + \frac{2\lambda(p/3)^2}{1 + (\cosh(2\lambda))^\gamma}\right) \quad (39)$$

$$\quad (40)$$

Justifications are:

(a) $D\left(f_G(\mathbf{x}^n) \| f_{G_1}(\mathbf{x}^n)\right) = nD\left(f_G(\mathbf{x}) \| f_{G_1}(\mathbf{x})\right)$ (due to independence of the $n$ samples) and $D\left(f_G(\mathbf{x}) \| f_{G_1}(\mathbf{x})\right) \leq \sum_{s,t \in V, s \neq t}\left(\theta_{s,t} - \theta'_{s,t}\right)\left(\mathbb{E}_G[x_s x_t] - \mathbb{E}_{G'}[x_s x_t]\right) \leq \lambda(2)\binom{p}{2}$. This is because there are $\binom{p}{2}$ edges and correlation is at most 1. Upper bound for $P((\mathcal{R}^p \epsilon)^c)$ is from Lemma 10. $G$ and $G_{\mathcal{D}=\emptyset}$ differ only in the edges present in $\mathcal{D}$ and irrespective of the edges in $\mathcal{D}$, all node pairs in $\mathcal{D}$ are $(2, \gamma)$ connected by definition of $\mathcal{D}$. Therefore, the second set of terms in (38) is bounded using Lemma 3.

Substituting the upper bound (39) in (27) and rearranging terms, we have the following lower bound for the number of samples needed when $c = \Omega(p^{3/4+\epsilon'})$, $\epsilon' > 0$:

$$n \geq \frac{\binom{p}{2}H(c/p)(1+\epsilon)}{\left(2\lambda\binom{p}{2}(2b_p + 2r_c) + \frac{2\lambda(p/3)^2}{1+(\cosh(2\lambda))^\gamma}\right)}\left(\frac{1}{1+\epsilon} - \frac{p_{\text{avg}}}{1-a_p} - \frac{\log|S|}{\binom{p}{2}H(c/p)(1+\epsilon)} - \frac{1}{\binom{p}{2}H(c/p)(1+\epsilon)}\right)$$

$$\overset{a}{\geq} \frac{H(c/p)(1+\epsilon)}{\left((4\lambda p/3)\exp(-(p/36) + 4\exp(-\frac{c^2\epsilon^2}{36})) + \frac{(4/9)\lambda}{1+(\cosh(2\lambda))^\gamma}\right)}\left(\frac{1}{10}\left(\frac{1-\frac{11}{9}\epsilon}{1+\epsilon}\right) - \frac{p_{\text{avg}}}{1-a_p} - O(1/p)\right)$$

$$\overset{\epsilon=1/2}{\geq} \frac{H(c/p)(3/2)}{\left((4\lambda p/3)\exp(-(p/36) + 4\exp(-\frac{c^2}{144})) + \frac{(4/9)\lambda}{1+(\cosh(2\lambda))^\gamma}\right)}\left(\frac{1}{40} - \frac{p_{\text{avg}}}{1-a_p} - O(1/p)\right)$$

$$\overset{\text{large } p}{\geq} \frac{H(c/p)(3/2)}{\left((4\lambda p/3)\exp(-(p/36) + 4\exp(-\frac{c^2}{144})) + \frac{(4/9)\lambda}{1+(\cosh(2\lambda))^\gamma}\right)}\left(\frac{1}{40} - 2p_{\text{avg}} - O(1/p)\right)$$

$$\overset{c=\Omega(p^{3/4}),\gamma=\frac{c^2}{6p}}{\geq} \frac{H(c/p)(3/2)}{\left((4\lambda p/3)\exp(-(p/36)) + 4\exp(-\frac{p^{\frac{3}{2}}}{144}) + \frac{(4/9)\lambda}{1+(\cosh(2\lambda))^{\frac{c^2}{6p}}}\right)}\frac{1}{40}(1 - 80p_{\text{avg}} - O(1/p))$$

$$\quad (41)$$

$$\quad (42)$$

(a)- This is obtained by substituting all the bounds for $b_p$ and $r_c$ and $\log|S|$ from Section 10.2.

From counting arguments in [1], we have the following lower bound for $G(p, c/p)$.

**Lemma 13.** *[1] Let $G \sim G(p, c/p)$. Then the average error $p_{\text{avg}}$ and the number of samples for this random graph class must satisfy:*

$$n \geq \frac{\binom{p}{2}}{p}H(c/p)\left(1 - \epsilon - p_{\text{avg}}(1+\epsilon)\right) - O(1/p) \quad (43)$$

*for any constant $\epsilon > 0$.*

Combining Lemma 13 with $\epsilon = 1/2$ and (41), we have the result in Theorem 6.