[Reviews · NeurIPS 2014]

Submitted by Assigned_Reviewer_7

This theoretical paper derives low bounds on the number of samples needed to learn graphs (and not factors) from data.
The paper uses various versions of the Fano lemma, e.g. conditional to particular graph classes. Derivations and arguments are purely information-theoretic. The results are probably correct (I did not check the algebra), however I am not sure about the value of what is reported. Here is the list of deficiencies/questions troubling me in this paper:
1) Assumption on the ferromagnetic and zero-field nature of the Ising model is highly unrealistic for majority of interesting cases.
2) Assumption on the constancy of the factors (leaving only an “effective temperature” in otherwise uniform ferromagnetic Ising model) within the graph makes it even more restrictive. I am not aware of any learning or inference application where such an assumption may be of a practical inference.
3) Once the (conditional) Fano lemma is formulated (and this part is rather straightforward and formal) the authors turn to finding subsets of sub-graphs which are large in size and yet have small KL-diameter. Question of the KL divergence is then linked to analysis of the difference in correlations between two representative graphs within the class. Next , the authors suggest that the difference between correlations will be small if one of the correlations is very large, that is when ``spins” at a pair of measurement points are completely aligned (very small temperature), or (another scenario) when compared graphs are simply close to each other. These two principles are then used in the ``application” part to modify a preselected candidate sub-graph (as a good candidate) in the way that the new sub-graph would remain close (in terms of the KL divergence). These arguments are correct but trivial – using very little information about the potentially rich correlations GM of the Ising type can provide. This observation emphasizes the very loose nature of the bounds derived.
4) In the application sections (#5,6) the authors apply the scheme to estimate the lower bounds for graphs of special types --- limited by path-length or girth and finally random Erdos-Renyi graphs. And here again, straightforward estimations (algebra used) seem correct, however I don’t really see a value in any of these results.
5) The story sold is not constructive. I did not find anywhere in the paper discussion of possible consequences of the information-theoretic consideration for actual learning algorithms.
Summary: Overall, I don’t see much of a value for NIPS community in publishing this paper describing theoretical bounds which are
(a) applied only to very limited class of models and
(b) based on simplistic arguments therefore (most probably) producing rather loose bounds
(c) not supported by simulations/experiments.

Submitted by Assigned_Reviewer_29

The paper considers the following interesting inverse problem: recover underlying graph of an Ising model from samples of the model. This is a well motivated problem (especially considering the future extensions to general graphical models). The closest related paper is [16] = "Information-Theoretic Limits of Selecting
Binary Graphical Models in High Dimensions (2012)". The setting in the paper under review is homogeneous case with known interaction (interaction on each edge is the same and the parameter (lambda) is known). The goal is to understand the number of samples (n) needed to have a small probability of misclassification (or expected probability of misclassification for graph families with a "prior"). The paper proves lower bounds on the number of samples needed.

The starting point is Corollary 2 (proved using Fano's lemma) which asks to identify a large subset of graphs with small eps-covering (under symmetric divergence). The distance (symmetric divergence) can be bounded (following the strategy from [16]) by differences in pairwise correlations; these are tractable when one has

high connectivity/large lambda - (correlations close to 1)

low connectivity/small lambda - (correlations close to 0)

The authors obtain a collection of lower bounds. Is is unclear how good (close to the upper bounds) the bounds are. Perhaps, the most interesting result is Corollary 4, where a threshold behavior (on Erdos-Renyi random graphs with very specific edge density) is identified.

Question: is there a connection between the hardness of the problem and phase transitions? (e.g., in Corollary 4 does the threshold match the uniqueness phase transition threshold?)
Summary: Overall, the question, techniques, and results are all interesting and non-trivial.

Submitted by Assigned_Reviewer_32

This paper provides a general framework for computing lower-bounds on the sample complexity of recovering the underlying graphs of Ising models, given i:i:d: samples. While there have been recent results for specific graph classes, these involve fairly extensive technical arguments that are specialized to each specific graph class. In contrast, this paper isolates two key graph-structural ingredients that can then be used to specify sample complexity lower-bounds. Presence of these structural properties makes the graph class hard to learn. This paper derives corollaries of the main result that not only recover existing recent results, but also provides lower bounds for novel graph classes not considered till now, extends our framework to the random graph setting, and derives corollaries for Erdos-Renyi graphs in a certain dense setting.

Quality: The quality of writing, discusssion, results, etc. is very high.

Clarity: The logic for discussion (Sections 3 ->4 -> 5 and 6) is beautiful.

Originality: Although using Fano inequality is standard, however, the authors considered a novel strategy: \Tau should be large and KL divergence should be small. To this end, they proved Lemmas 3,4, and Corollary 3. Although the detailed strategies in Section 5 can be applied to each individual situations, the lower counds are successfully obtained.

Significance: The fact that the authors obtained those lower bounds on sample complexity is not a big news. However, the general strategy proposed by this paper would be followed by many researchers in the future.
Summary: I took many hours to read the manuscript but was impressed finally. This paper should be accepted.
Author Feedback
Author rebuttal: We would like to the thank the reviewers for their careful reviews and comments.

Reviewer 1
------------

1. Regarding connection to phase transitions:

That is certainly a very interesting question. For some explicit algorithms, this has been considered in some previous work (Ref: "Which graphical models are difficult to learn?" Bento and Montanari, 2009) wherein they show for certain algorithms and the degree bounded ensemble, that while having \lambda < const./maxdeg allows for bounded sample complexity, but if we have \lambda > const./maxdeg, then the algorithm may require potentially infinite samples (i.e. the algorithm fails w.h.p.).

When d tanh(\lambda) < 1 (roughly lambda< const./d) mixing time of glauber dynamics is polynomial for the ensemble of degree d-bounded graphs and other wise it is exponential. For Erdos-Renyi graphs G(p,d/p), the phase transition occurs at d tanh(\lambda) <1 (see Ref: "Exact thresholds for Ising-Gibbs samplers on general graphs", Mossel and Sly 2013). For degree d-bounded graphs, the uniqueness threshold (for spatial mixing) is identical to the threshold for glauber dynamics.

Through information-theoretic lower bounds, we can get a similar scaling for the lambda threshold for the degree bounded case if we want to avoid an exponential growth in the number of samples to learn. For Erdos-Renyi graphs, our learning threshold does not match the mixing time threshold except when d=O(p). For Erdos-Renyi graphs, there is a possibility of tightening our results and further extending it to other ranges of parameters (range of average degree d). At this point, connection between learning threshold and the mixing thresholds is not clear to us.

Our bounds allow us to obtain restrictions on edge weights for other graph ensembles, for which these connections are unexplored. Another interesting line of research is to consider computation in tandem with information-theoretic restrictions to obtain bounds.

Reviewer 2
------------

Thank you for your kind comments and feedback.

Reviewer 3
------------

1. Regarding the use of model restrictions (ferromagnetic, zero-field, constant weights):

Note that lower bounds for a set of Ising models with restrictions *are also applicable* to the set of all Ising models without these restrictions (as we noted briefly in Footnote 1, page 3).

As regards the specific choice of restrictions, these were simply used for the construction of the motifs considered in this paper. Of course, it may be possible to construct motifs without these restrictions. Nevertheless, in many of the use-cases in the paper, these bounds are certainly tight --- this includes the degree-restricted graphs where they match an upper bound from known algorithms (making these tight), and also the path-restricted graphs which again match an upper bound from a known algorithm under certain conditions.

2. Regarding the arguments being "correct but trivial":

We attempted to keep the arguments as simple as possible, but as mentioned in the preceding paragraph, these certainly seem to be good for many deterministic scenarios considered previously. We note that our arguments for the case of random (e.g. Erdos-Renyi) graphs in particular however are highly non-trivial: it is not enough to just identify a subset of graphs that have a low K-divergence between each other. Since average probability of error needs to be bounded in a random ensemble, one needs to show that there exists a very small covering set of graphs where many pairs of nodes are highly connected and the this small set 'covers' almost all the graphs (in measure) in the sense that any graph differs only in the well connected pairs from some graph in the covering set. This is a general principle that has been outlined in Sec 6.1 that could be potentially used for other random ensembles like power law graphs. This approach identifies the scaling of lambda necessary for the average error probability to be low. As far as we are aware, such a threshold computation for weights for average error probability has not been done for a random ensemble of GMs before.

3. Regarding the value of our results and lower bounds in general:

The key value of lower bounds is that they can serve as certificates of optimality for candidate algorithms (so long as their sample complexities achieve these lower bounds), or as a spur to develop such optimal algorithms. Beyond these consequences, as we pointed out in the application to specific classes, the lower bounds show the kind of restrictions on the edge weights \lambda, that would be necessary if the algorithm wants to avoid exponential sample complexity requirements. We would also like to refer the reviewer to the previous reply for possible consequences of the lower bounding approach being applied to studying power law GMs of which Erdos Renyi graphs is a special case.

4. Regarding Simulations/Experiments:

We have not proposed any explicit algorithm to meet all these lower bounds universally. However, algorithms previously proposed for certain deterministic classes match our lower bounds. Further, we have clarified regimes for parameters (the weight parameter lambda) in which any experiment/simulation based on any algorithm would require prohibitively large sample complexity.